# Cooperative interactions facilitate stimulation of Rad51 by the Swi5-Sfr1 auxiliary factor complex

Bilge Argunhan[1†], Masayoshi Sakakura[2†], Negar Afshar[1,3], Misato Kurihara[2‡], Kentaro Ito[1], Takahisa Maki[1], Shuji Kanamaru[1,3], Yasuto Murayama[4], Hideo Tsubouchi[1,3], Masayuki Takahashi[3], Hideo Takahashi[2]*, Hiroshi Iwasaki[1,3]*

[1]Institute of Innovative Research, Tokyo Institute of Technology, Tokyo, Japan; [2]Graduate School of Medical Life Science, Yokohama City University, Yokohama, Japan; [3]School of Life Science and Technology, Tokyo Institute of Technology, Tokyo, Japan; [4]Center for Frontier Research, National Institute of Genetics, Shizuoka, Japan

**Abstract** Although Rad51 is the key protein in homologous recombination (HR), a major DNA double-strand break repair pathway, several auxiliary factors interact with Rad51 to promote productive HR. We present an interdisciplinary characterization of the interaction between Rad51 and Swi5-Sfr1, a conserved auxiliary factor. Two distinct sites within the intrinsically disordered N-terminus of Sfr1 (Sfr1N) were found to cooperatively bind Rad51. Deletion of this domain impaired Rad51 stimulation in vitro and rendered cells sensitive to DNA damage. By contrast, amino acid-substitution mutants, which had comparable biochemical defects, could promote DNA repair, suggesting that Sfr1N has another role in addition to Rad51 binding. Unexpectedly, the DNA repair observed in these mutants was dependent on Rad55-Rad57, another auxiliary factor complex hitherto thought to function independently of Swi5-Sfr1. When combined with the finding that they form a higher-order complex, our results imply that Swi5-Sfr1 and Rad55-Rad57 can collaboratively stimulate Rad51 in *Schizosaccharomyces pombe*.

*For correspondence:
hidtak@yokohama-cu.ac.jp (HT);
hiwasaki@bio.titech.ac.jp (HI)

†These authors contributed equally to this work

Present address: ‡Nomura Research Institute Ltd., Tokyo, Japan

Competing interests: The authors declare that no competing interests exist.

## Introduction

DNA double-strand breaks (DSBs) are a particularly toxic form of DNA damage in which a DNA molecule is broken into two (or more) fragments. A major DSB repair pathway is homologous recombination (HR). During HR, an intact stretch of DNA that shares sequence similarity to the DSB site is identified and utilized as a template for synthesis-dependent repair. Dysregulation of HR results in misrepair of DSBs, resulting in genomic instability, a potent driver of tumorigenesis (*Jeggo et al., 2016*).

HR is initiated by the formation of 3' single-stranded DNA (ssDNA) at the DSB site. This ssDNA is first bound by RPA, then by the ubiquitous RecA-family recombinase Rad51, which forms a right-handed nucleoprotein filament. The Rad51 filament is able to capture intact double-stranded DNA (dsDNA) and—by assessing the extent of base-pairing with the filamentous ssDNA—identify regions of DNA that share substantial sequence similarity to the DSB site (*Prentiss et al., 2015*). After initial pairing with the complementary strand of the dsDNA, the Rad51 filament further displaces the non-complementary strand by driving strand transfer, resulting in the formation of an intermediate structure known as a displacement loop. The 3' end of the invading strand is then utilized as a primer for DNA synthesis, leading to its extension and the recovery of lost genetic information. In the simplest case, ejection of this extended strand allows it to anneal with the complementary DNA on the other

**eLife digest** The DNA within cells contains the instructions necessary for life and it must be carefully maintained. DNA is constantly being damaged by radiation and other factors so cells have evolved an arsenal of mechanisms that repair this damage. An enzyme called Rad51 drives one such DNA repair process known as homologous recombination.

A pair of regulatory proteins known as the Swi5-Sfr1 complex binds to Rad51 and activates it. The complex can be thought of as containing two modules with distinct roles: one comprising the first half of the Sfr1 protein and that is capable of binding to Rad51, and a second consisting of the rest of Sfr1 bound to Swi5, which is responsible for activating Rad51. Here, Argunhan, Sakakura et al. used genetic and biochemical approaches to study how this first module, known as "Sfr1N", interacts with Rad51 in a microbe known as fission yeast.

The experiments showed that both modules of Swi5-Sfr1 were important for Rad51 to drive homologous recombination. Swi5-Sfr1 complexes carrying mutations in the region of Sfr1N that binds to Rad51 were unable to activate Rad51 in a test tube. However, fission yeast cells containing the same mutations were able to repair their DNA without problems. This was due to the presence of another pair of proteins known as the Rad55-Rad57 complex that also bound to Swi5-Sfr1.

The findings of Argunhan, Sakakura et al. suggest that the Swi5-Sfr1 and Rad55-Rad57 complexes work together to activate Rad51. Many genetically inherited diseases and cancers have been linked to mutations in DNA repair proteins. The fundamental mechanisms of DNA repair are very similar from yeast to humans and other animals, therefore, understanding the details of DNA repair in yeast may ultimately benefit human health in the future.

side of the DSB (*Mehta and Haber, 2014*). Following gap filling by further DNA synthesis and ligation of resultant nicks, recombinational DNA repair is complete.

Efforts to elucidate the underlying biochemistry of HR have typically involved measuring the ability of purified Rad51 to drive pairing and subsequent strand transfer of homologous DNA substrates, a process known as DNA strand exchange. Such experiments established RPA as a critical component of the DNA strand exchange reaction (*Sung, 1994*). However, when RPA was added to the reaction concomitantly with Rad51, which more closely reflects the situation in vivo, the stimulatory effect of RPA was abolished. This paradoxical finding led to the discovery that other proteins known to be involved in HR serve as auxiliary factors that interact directly with Rad51 and can negate the inhibitory effect of RPA (*Hays et al., 1995*; *Johnson and Symington, 1995*; *Sung, 1997a*; *Sung, 1997b*; *Tsutsui et al., 2000*; *Tsutsui et al., 2001*; *Jensen et al., 2010*).

Numerous distinct families of recombination auxiliary factors have been identified throughout eukaryotes, including Rad52, BRCA2, Rad54, Rad51 paralogs, Swi5-Sfr1 and the Shu complex (*Zelensky et al., 2014*). Each group is thought to have non-overlapping roles in HR, although the mechanistic differences are yet to be elucidated (*Akamatsu et al., 2007*; *Khasanov et al., 2004*; *Martín et al., 2006*; *Shor et al., 2005*; *Sugawara et al., 2003*). Aside from the Shu complex, all auxiliary factors are capable of binding directly to Rad51, which is thought to be essential for their respective roles in HR (*Zelensky et al., 2014*). Sfr1 was first discovered in the fission yeast *Schizosaccharomyces pombe* as an interactor of Rad51, and along with Swi5, was shown to comprise an HR sub-pathway that functions independently of and in parallel to the Rad51 paralogs Rad55-Rad57 (*Akamatsu et al., 2003*; *Akamatsu et al., 2007*). Subsequent biochemical reconstitutions demonstrated that substoichiometric concentrations of Swi5-Sfr1 were able to efficiently stimulate the strand exchange activity of Rad51 and Dmc1, the meiosis-specific RecA-family recombinase (*Haruta et al., 2006*). This enhancement of strand exchange was attributed to stabilization of the nucleoprotein filaments and stimulation of the recombinases' ATPase activity (*Haruta et al., 2006*; *Kurokawa et al., 2008*; *Murayama et al., 2013*). Rad51-driven DNA strand exchange was recently shown to fit a three-step kinetic model with two reaction intermediates (*Ito et al., 2018*). Swi5-Sfr1 enhanced transitioning of the first three-strand intermediate (corresponding to a paranemic joint) into the second three-strand intermediate (corresponding to a plectonemic joint), and conversion of the second intermediate into reaction products, thus making it the only auxiliary factor known to

potentiate Rad51 in both the presynaptic and synaptic phases of DNA strand exchange. These findings highlight the unique role of Swi5-Sfr1 as an HR regulator.

Limited proteolysis of Swi5-Sfr1 yielded a stable C-terminal Sfr1 fragment in complex with Swi5 (Swi5-Sfr1C, residues 181–299 of Sfr1; Kuwabara et al., 2010), and this, along with the N-terminal half of Sfr1 (Sfr1N, residues 1–176 of Sfr1), could be stably expressed and purified (Kuwabara et al., 2012). Whereas Sfr1N was predicted to be intrinsically disordered (Kokabu et al., 2011; Saikusa et al., 2013), crystallographic analyses demonstrated that Swi5-Sfr1C forms a kinked structure that is able to stimulate Rad51-driven strand exchange by stabilizing the presynaptic filament and enhancing the ATPase activity of Rad51 (Kuwabara et al., 2012). By contrast, Sfr1N had no direct effect on these activities but was seen to co-immunoprecipitate (co-IP) with Rad51. Such complex formation was not detected between Rad51 and Swi5-Sfr1C, despite the stimulatory effect of Swi5-Sfr1C on Rad51. Taken together with the observation that Swi5-Sfr1C was only able to stimulate Rad51 activity when present at much higher concentrations than full-length Swi5-Sfr1, these results led to a model in which Sfr1N keeps Swi5-Sfr1C anchored in close proximity to Rad51 (Kuwabara et al., 2012).

Due to the use of truncated proteins in which entire domains were deleted (Kuwabara et al., 2012), it was not possible to determine whether Sfr1N has any function other than anchoring Swi5-Sfr1 to Rad51. To explore this, we employed an interdisciplinary approach to further characterize Sfr1N. We provide direct evidence that Sfr1N is intrinsically disordered and contains two sites that interact cooperatively with Rad51. Mutation of critical residues within these two sites rendered Rad51 refractory to the stimulatory effects of full-length Swi5-Sfr1, mimicking the results obtained with Swi5-Sfr1C (i.e., when the N-terminus of Sfr1 is absent), indicating that the primary function of Sfr1N is to facilitate the interaction between Swi5-Sfr1 with Rad51. Unexpectedly, and in contrast to the severely impaired Rad51 stimulation observed in vitro, these interaction mutants only showed defects in Rad51-mediated DNA repair in the absence of Rad55-Rad57, implying that these Rad51 paralogs can facilitate Swi5-Sfr1-dependent DNA repair. Consistent with this possibility, purified Swi5-Sfr1 was found to interact with partially purified Rad55-Rad57. Collectively, these results provide a molecular basis for Rad51 stimulation by Swi5-Sfr1 and reveal a novel interplay between recombination auxiliary factors.

## Results

### Sfr1N is essential for the role of Swi5-Sfr1 in DNA repair

Since Sfr1N binds to Rad51 but does not stimulate DNA strand exchange, and Swi5-Sfr1C stimulates DNA strand exchange despite not forming a detectable complex with Rad51, it was proposed that Sfr1N functions exclusively to facilitate the interaction between Swi5-Sfr1C and Rad51 (Kuwabara et al., 2012). However, it remained possible that Sfr1N only exerts a stimulatory effect when in the presence of Swi5-Sfr1C. To test this, strand exchange reactions containing purified Rad51 and plasmid-sized DNA substrates (Figure 1A) were supplemented with equimolar concentrations of both Sfr1N and Swi5-Sfr1C. Even in this setting, Sfr1N did not have any stimulatory effect on DNA strand exchange (Figure 1B,C), raising the possibility that it is dispensable for the physiological function of Swi5-Sfr1. To determine the requirement for these two modules in Rad51-dependent DNA repair, strains lacking the C-terminal or N-terminal half of Sfr1 (sfr1N and sfr1C, respectively) were constructed. Both strains showed the same sensitivity to DNA damage as a strain in which Sfr1 was completely absent (sfr1Δ; Figure 1D). Furthermore, combining these truncations with rad55Δ sensitized cells to DNA damaging agents to the same degree as the rad55Δ sfr1Δ strain, which displays a complete loss of Rad51-dependent DNA repair (Figure 1—figure supplement 1A; Akamatsu et al., 2003; Akamatsu et al., 2007). Sfr1N and Sfr1C were detected at comparable levels to full-length Sfr1 by immunoblotting, indicating that the sensitivity of the sfr1N and sfr1C strains is not due to a reduction in protein levels (Figure 1—figure supplement 1B). Furthermore, this sensitivity was not rescued by fusing Sfr1N or Sfr1C to the SV40 large T antigen nuclear localization signal, suggesting that the observed phenotype is not caused by a failure to localize to the nucleus (Figure 1—figure supplement 1C). Thus, although not essential for stimulation of Rad51 in vitro, Sfr1N is essential for the function of Swi5-Sfr1 in promoting Rad51-dependent DNA repair.

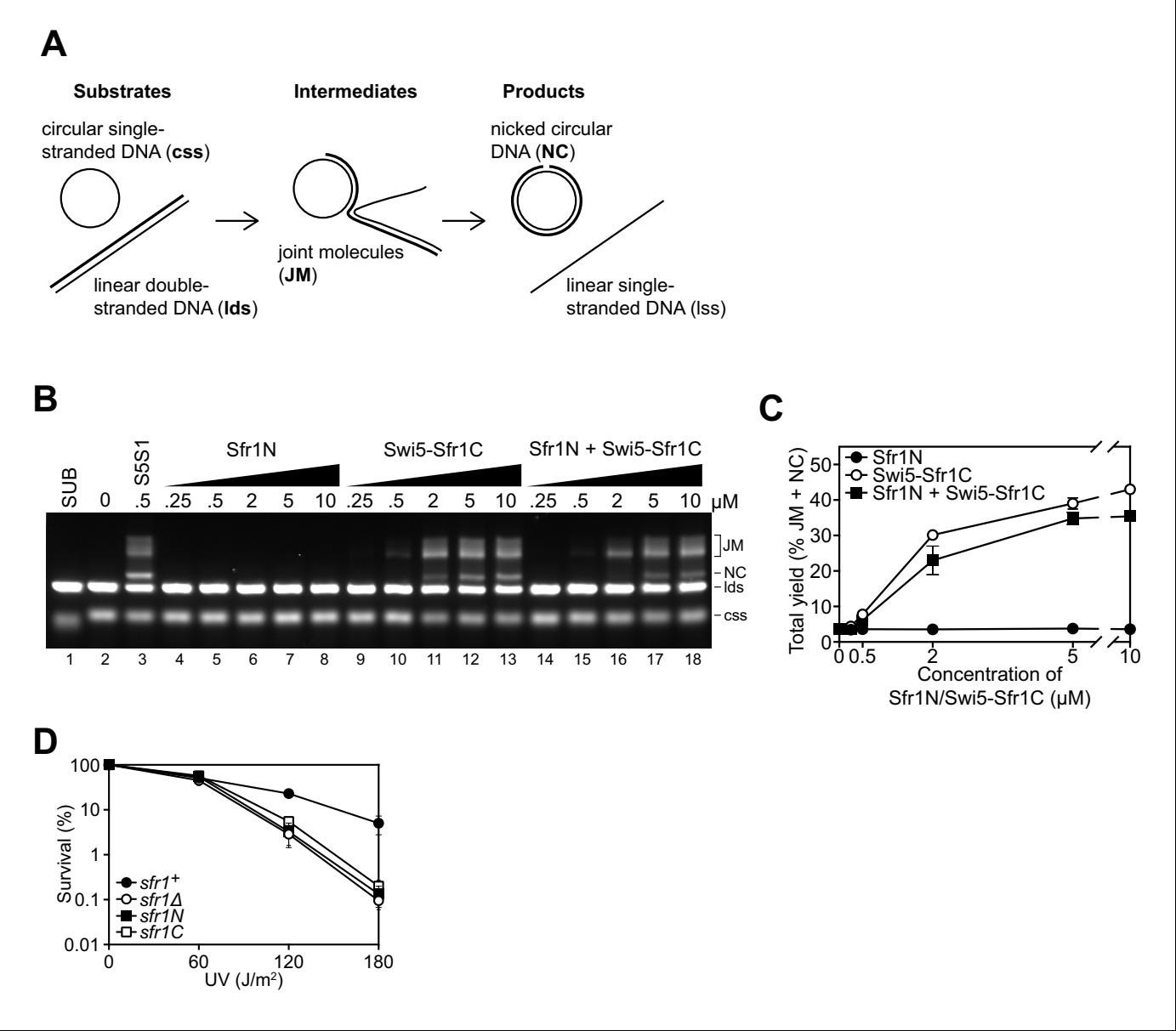

**Figure 1.** Sfr1N is essential for DNA repair mediated by Swi5-Sfr1. (**A**) Schematic of the three-strand exchange assay. (**B**) Three-strand exchange reactions containing Rad51 with the indicated concentrations of Swi5-Sfr1, or Sfr1N and/or Swi5-Sfr1C, were incubated for 2 hr at 37 °C and DNA was separated by agarose gel electrophoresis to visualize substrates (css, lds), intermediates (JM) and products (NC). (**C**) The percentage of DNA signal per lane corresponding to JM and NC (total yield) was plotted against protein concentration. (**D**) Percentage of cell survival following acute UV irradiation for the indicated strains. For (**C,D**), mean values of three independent experiments ± s.d. are shown.

The online version of this article includes the following source data and figure supplement(s) for figure 1:

**Source data 1.** Yield (%) of total DNA for data in *Figure 1C*.
**Figure supplement 1.** The DNA damage sensitivity of *sfr1N/C* is not due to protein instability and is not suppressed by fusion to an NLS.

## Sfr1N comprises an intrinsically disordered and flexible domain within the Swi5-Sfr1 ensemble

Having confirmed the physiological importance of Sfr1N, a structural approach was employed to glean insights into its molecular function. Primary sequence analysis and ion mobility mass spectrometry of Sfr1N suggested that this domain is intrinsically disordered (*Kokabu et al., 2011*; *Saikusa et al., 2013*). To directly test this, Sfr1N was analyzed by circular dichroism (CD) and nuclear magnetic resonance (NMR) spectroscopy. The far-UV CD spectrum of Sfr1N lacked local minima

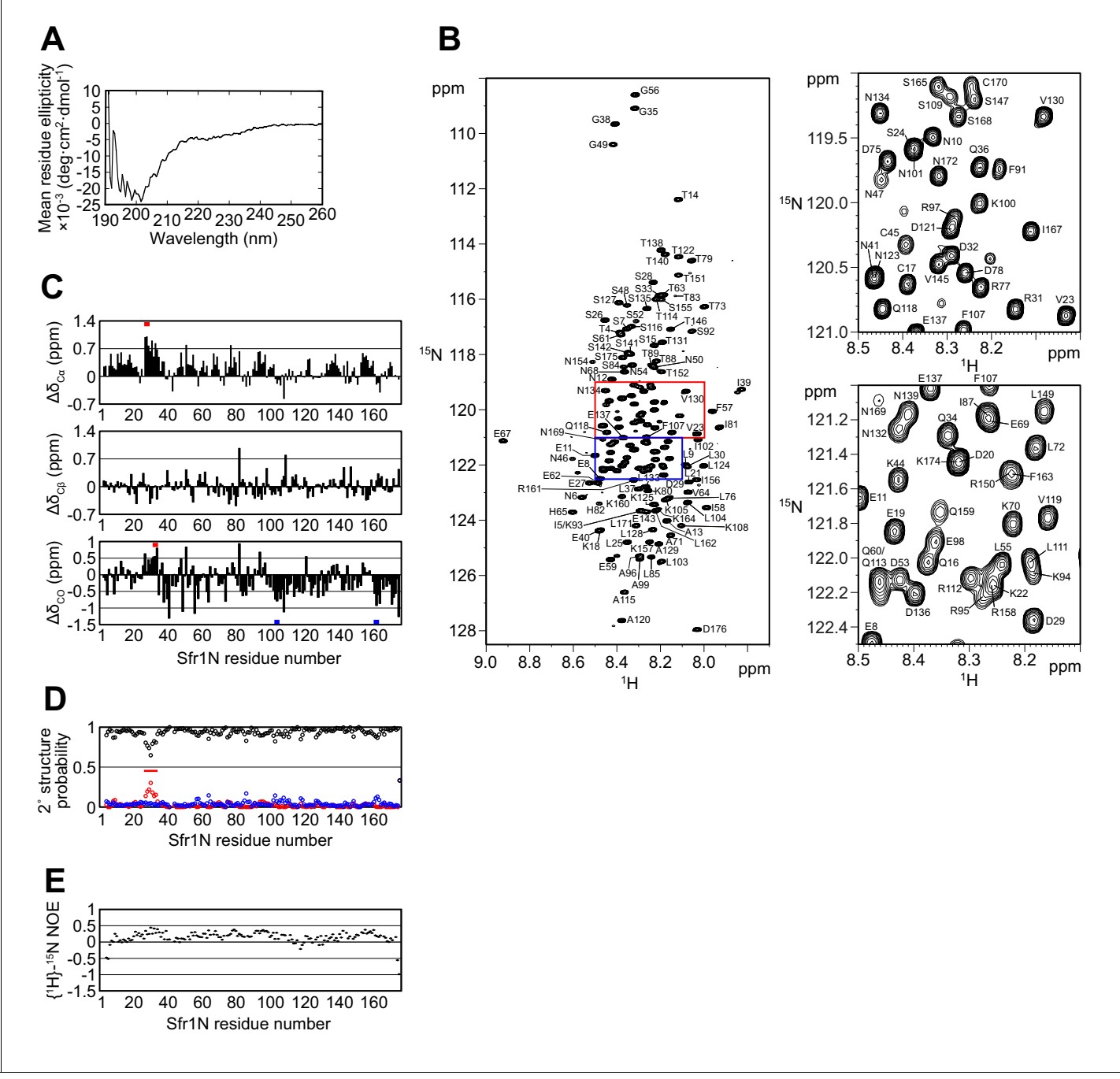

**Figure 2.** The N-terminal half of Sfr1 is an intrinsically disordered and flexible domain. (A) Far-UV CD spectrum of Sfr1N. (B) $^{1}$H-$^{15}$N HSQC spectrum of Sfr1N. Regions outlined in red and blue are enlarged in the top right and bottom right panels, respectively. (C) Secondary chemical shifts of $^{13}$C$_{\alpha}$ (top), $^{13}$C$_{\beta}$ (middle), and $^{13}$CO (bottom) were obtained and plotted against the corresponding position in Sfr1N. Values that suggest potential α-helix and β-strand formation are indicated with red and blue bars, respectively. (D) TALOS+ prediction of secondary structure probabilities. Black, red, and blue circles indicate the probabilities of random coils, α-helices, and β-strands, respectively. (E) {$^{1}$H}-$^{15}$N heteronuclear NOE was measured for Sfr1N. All analyzed main chain NHs showed NOE values of less than 0.44, with a protein-wide average of 0.15.

The online version of this article includes the following figure supplement(s) for figure 2:

**Figure supplement 1.** Validation of Sfr1N signals by $^{15}$N labeling.

**Figure supplement 2.** N-terminal intrinsically disordered domains may be an evolutionarily conserved feature of Sfr1.

above 210 nm and showed a negative peak at ~200 nm (*Figure 2A*), implying a lack of secondary structural units such as α-helices and β-sheets (*Greenfield, 2006*). Furthermore, examination of the $^1H$-$^{15}N$ heteronuclear single quantum coherence (HSQC) spectrum of Sfr1N revealed that most of the main chain amide protons resonated in a narrow chemical shift range between 7.7 and 8.7 ppm (*Figure 2B*), which is a characteristic feature of disordered proteins (*Konrat, 2014*). To extract structural information for each residue, NMR signals from main-chain $^1H_N$, $^{13}C_\alpha$, $^{13}CO$, and $^{15}N_H$ atoms as well as $^{13}C_\beta$ resonances were assigned by analyzing a set of triple resonance spectra. This was assisted by the $^1H$-$^{15}N$ HSQC spectra of selectively-$^{15}N$ labeled versions of Sfr1N and several Sfr1N variants (*Figure 2—figure supplement 1*).

The chemical shifts obtained for the main-chain and $^{13}C_\beta$ atoms enabled secondary structure prediction. The secondary chemical shift of $^{13}C_\alpha$, $^{13}C_\beta$, and $^{13}CO$ atoms, which is the chemical shift difference between the measured values and the corresponding amino acids in random coil peptides,

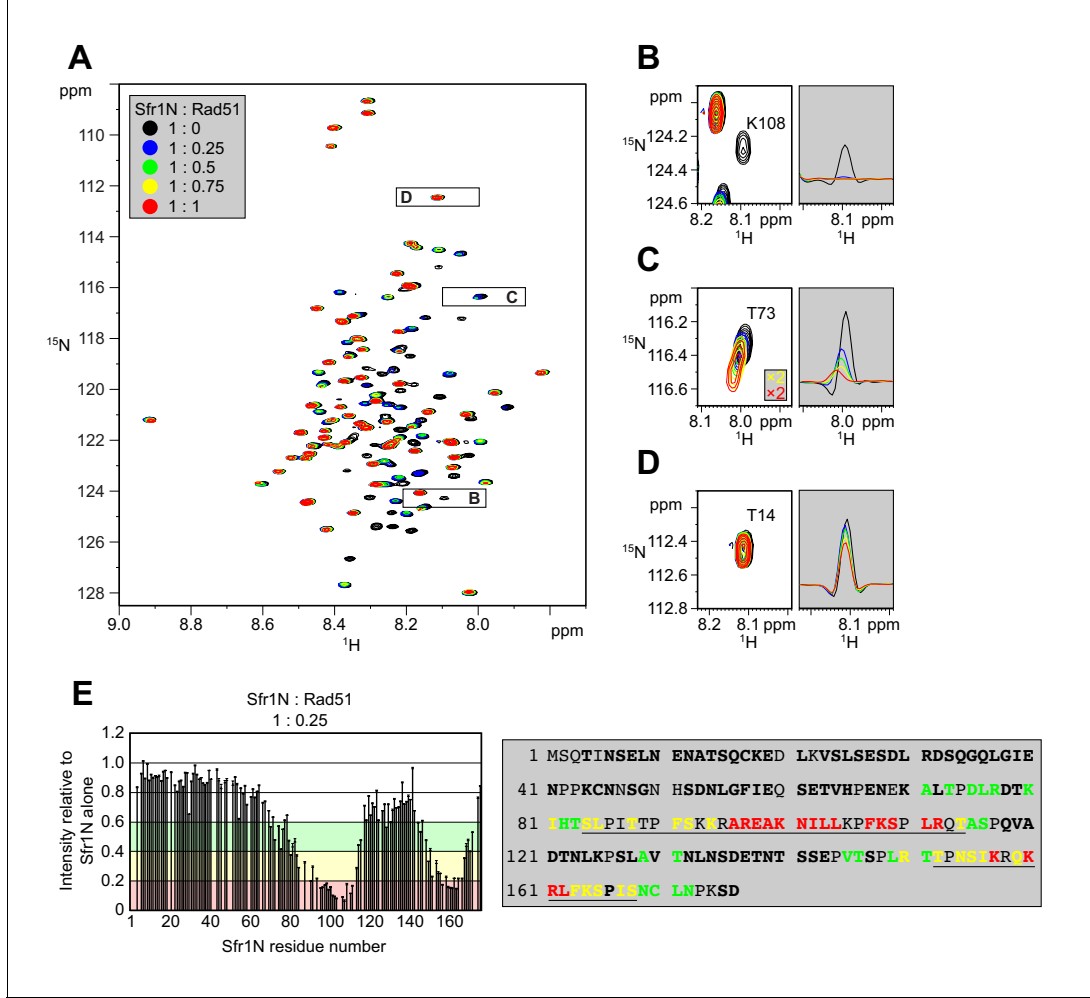

**Figure 3.** Two sites within Sfr1N interact with Rad51. (A) Superimposed $^1H$-$^{15}N$ HSQC spectra of $^{15}N$-labeled Sfr1N in the absence of Rad51 or in the presence of Rad51 at the indicated molecular ratios. (B–D) Enlarged signals from K108, T73, and T14 (left), with slices along the $^1H$ axis (right). For (C), the threshold of the yellow and red spectra was set 2x lower than the other spectra to show the signals clearly. (E) Signal intensity ratio of Sfr1N residues in the presence of Rad51 to those in the absence of Rad51 (left) and features of the Sfr1N amino acid sequence (right). Reductions in signal intensity of 40–60% (green), 60–80% (yellow), and >80% (red) are highlighted, along with the corresponding residues. Underlined residues correspond to Site 1 (S84 to T114) and Site 2 (T152 to S168), where the most significant signal attenuations were observed. Residues not in bold include prolines, unassigned residues, and residues with overlapped signals.

The online version of this article includes the following figure supplement(s) for figure 3:

**Figure supplement 1.** Changes in Sfr1N NMR signal intensity in response to Rad51 titration.

**Figure supplement 2.** Molecular sizing of *S. pombe* Rad51.

was determined (*Figure 2C*). The majority of Sfr1N residues showed $^{13}$C secondary chemical shift values within a limited range, suggesting that random coil structures are present in these regions. Nonetheless, a few groups of residues exhibited more than three consecutive secondary chemical shift values outside of this range, raising the possibility that some structures resembling α-helices (E27 to D29, D32 to Q34) or β-strands (L103 to K105, R161 to K164) may form within Sfr1N (*Wishart and Sykes, 1994*). Further secondary structure analysis was performed using the $^{1}$H$_N$, $^{13}$C$_\alpha$, $^{13}$C$_\beta$, $^{13}$CO, and $^{15}$N$_H$ chemical shift values and the program TALOS+ (*Shen et al., 2009*), which predicted Sfr1N to be entirely disordered, with a low probability for α-helix formation from E27 to S33 (*Figure 2D*).

To analyze the dynamical features of Sfr1N, the steady state heteronuclear nuclear Overhauser effect (NOE) for the main-chain amide groups of the protein was analyzed (*Farrow et al., 1994*; *Kay et al., 1989*). NOE values for all residues was less than 0.44, indicating that the entire protein is flexible with pico-to-nanosecond timescale motions (*Figure 2E*). Such fast motions are typically observed for unstructured proteins. The NOE values were not completely uniform. While the average NOE values for all Sfr1N residues was 0.15, residues E27 to S33 showed slightly increased values with an average of 0.32. Other regions within Sfr1N also showed similarly increased NOE values (e. g., residues F91 to A99, average of 0.32). However, unlike these other regions, several additional lines of evidence pointed toward the possibility that residues E27 to S33 may form an α-helix (see above). Collectively, these results demonstrate that, unlike the structured Swi5-Sfr1C complex (*Kuwabara et al., 2012*), the N-terminal half of Sfr1 is intrinsically disordered and flexible. Disorder predictions using DISOPRED3 (*Jones and Cozzetto, 2015*) suggest that the disordered state of Sfr1N may have been conserved throughout evolution (*Figure 2—figure supplement 2*, see Discussion for more details).

## Two sites within Sfr1N interact with Rad51

*Kuwabara et al. (2012)* suggested that Sfr1N facilitates the interaction between Swi5-Sfr1 and Rad51. To identify the site(s) within Sfr1N that binds to Rad51, NMR spectra of $^{15}$N-labeled Sfr1N were analyzed in the absence and presence of Rad51. Superimposed $^{1}$H-$^{15}$N HSQC spectra of $^{15}$N-labeled Sfr1N with increasing amounts of Rad51 were constructed (*Figure 3A*). The most prominent spectral changes, defined as a reduction in signal intensity of >80% (Sfr1N:Rad51 ratio of 1:0.25), were observed for 19 out of 142 non-overlapped residues (*Figure 3B,E*). In addition to these marked changes, 18 and 20 residues experienced signal intensity reductions of 60–80% and 40–60%, respectively (*Figure 3E*). The signal intensity of these residues was further attenuated upon incremental addition of Rad51 (*Figure 3—figure supplement 1A–C*). Most of these attenuated signals did not display obvious chemical shift changes following Rad51 binding. However, five residues (A71, T73, D75, L76, and T146) displayed incremental chemical shift changes and reductions in signal intensity upon addition of increasing amounts of Rad51 (*Figure 3C*, *Figure 3—figure supplement 1D*). The remaining 60 residues that were analyzed experienced minimal effects upon addition of Rad51 (<40% reduction in signal intensity; *Figure 3D,E*). These findings implicate two sites in Sfr1N, Site 1 (S84 to T114) and Site 2 (T152 to S168), where the most significantly attenuated signal intensities are sandwiched by moderately attenuated signal intensities (*Figure 3E*), as being important for the Sfr1N-Rad51 interaction. Site 1 is highly basic and hydrophobic compared to other regions of Sfr1N. Positively charged residues are also a prominent feature of Site 2, but this site is not especially hydrophobic. These results suggest that, while both Sites 1 and 2 are involved in electrostatic interactions with Rad51, Site 1 may also participate in hydrophobic interactions with Rad51. We note that, like all RecA-family recombinases, *S. pombe* Rad51 exists as a multimer in solution, with a size corresponding to ~160 kDa (*Figure 3—figure supplement 2*, see Discussion for more details).

To provide further support that Sites 1 and 2 within Sfr1N interact with Rad51, site-specific cross-linking experiments were conducted (*Figure 4A*). By utilizing *Escherichia coli* with an expanded genetic code, synthetic amino acids can be introduced at a site of interest via suppression of the amber (UAG) stop codon (*Young et al., 2010*). Translation is terminated prematurely without amber suppression (e.g., when the synthetic amino acid is omitted from the growth media), hence ensuring that all full-length protein products contain the synthetic amino acid. Several residues within Sites 1 and 2 were replaced with the photoreactive amino acid *p*-benzoyl-L-phenylalanine (*p*BPA). Following exposure to UV light, proteins within ~3 Å of *p*BPA become crosslinked to it (*Tanaka et al., 2008*). Such crosslinked proteins can be detected as slow-migrating species by immunoblotting and are

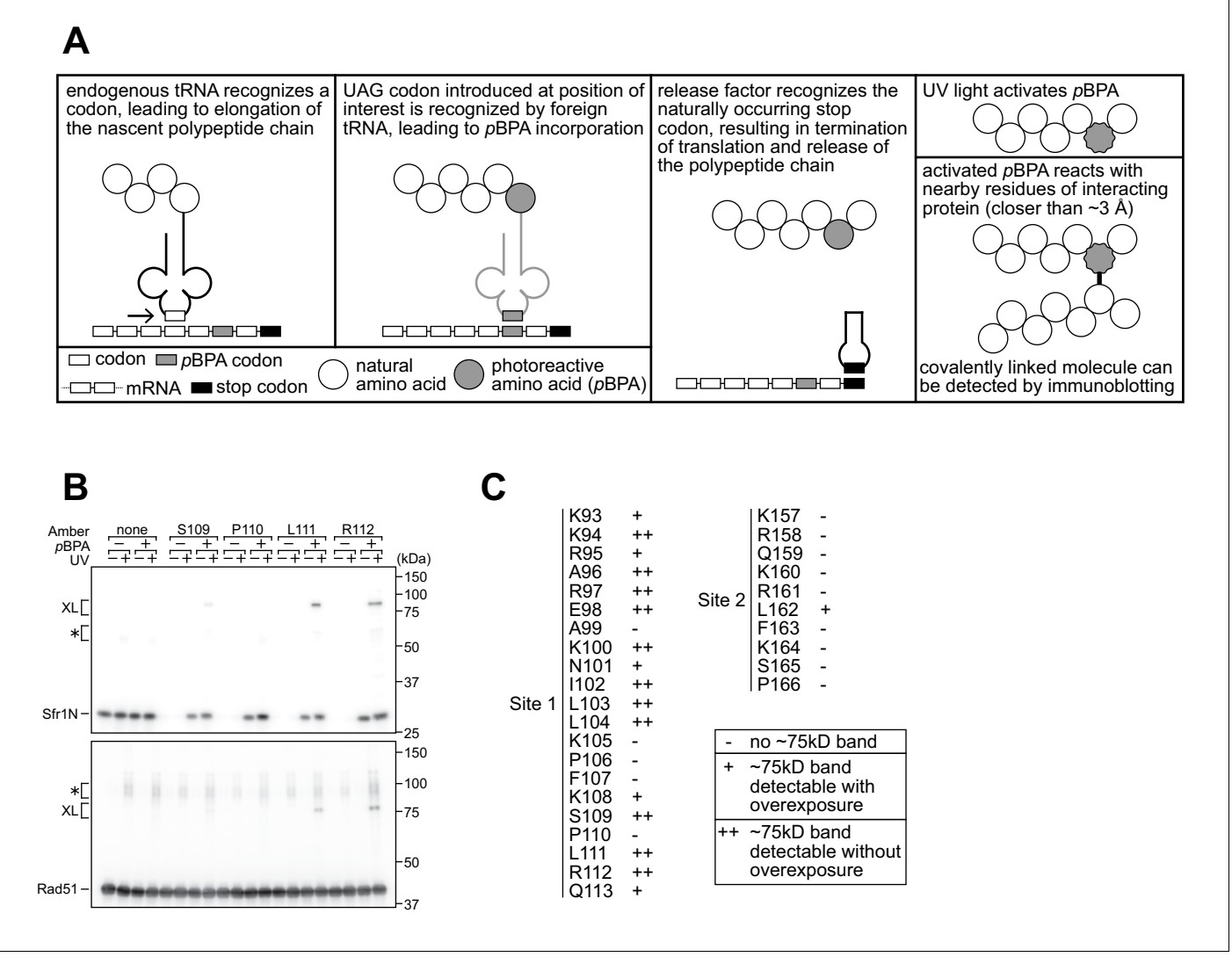

**Figure 4.** Residues within sites 1 and 2 can be site-specifically crosslinked to Rad51. (**A**) Schematic of the crosslinking assay. (**B**) Example of Sfr1N-Rad51 crosslinking. The indicated residues were replaced with an amber codon, or the amber codon was omitted as a negative control ('none'). XL, specific crosslinks (defined in the text). *non-specific crosslinks. (**C**) Summary of crosslinking results for all residues examined.

The online version of this article includes the following figure supplement(s) for figure 4:

**Figure supplement 1.** Nonspecific and specific crosslinking of Sfr1N to Rad51.

implicated in forming part of the interface in a protein-protein interaction (*Miyazaki et al., 2016*). Rad51 was co-expressed in *E. coli* with Sfr1N and cells were irradiated with UV. Proteins were then analyzed by immunoblotting with anti-Rad51 and anti-Sfr1 antibodies. Some non-specific crosslinking was observed when cells were treated with UV (*Figure 4—figure supplement 1A*). In addition to these non-specific crosslinks, numerous instances of specific crosslinking—defined as being dependent on a TAG mutation, the inclusion of *p*BPA in the media, and UV treatment—were observed (*Figure 4B*). Whereas several positions within Site 1 showed robust crosslinking to Rad51, positions within Site 2 showed little crosslinking (*Figure 4C*, *Figure 4—figure supplement 1B*), perhaps because the associations between Site 2 and Rad51 are more transient than those involving Site 1 (see Discussion). Taken together, the results obtained from the NMR interaction analysis and Site-specific crosslinking experiments indicate that two sites within Sfr1N, designated as Sites 1 and 2, interact with Rad51.

## Sites 1 and 2 cooperatively facilitate the physical and functional interaction between Swi5-Sfr1 and Rad51

A sequence alignment of Sfr1 orthologs within the genus *Schizosaccharomyces* highlighted conserved patches of positively charged residues in Sites 1 and 2 (bold residues in blue typeface, *Figure 5—figure supplement 1A*). Combined with the knowledge that Sfr1N only co-IPs with Rad51 under low-salt conditions (*Kuwabara et al., 2012*), it seemed plausible that these residues might be important for electrostatic interactions with Rad51. Hence, three residues in Site 1 were mutated (3A) and four residues in Site 2 were mutated (4A). Additionally, these mutants were combined to generate the 7A mutant (*Figure 5A*).

To directly assess whether these mutations disrupt the interaction with Rad51, Swi5 and full-length Sfr1 were co-purified to homogeneity (*Figure 5—figure supplement 1B*). Next, purified

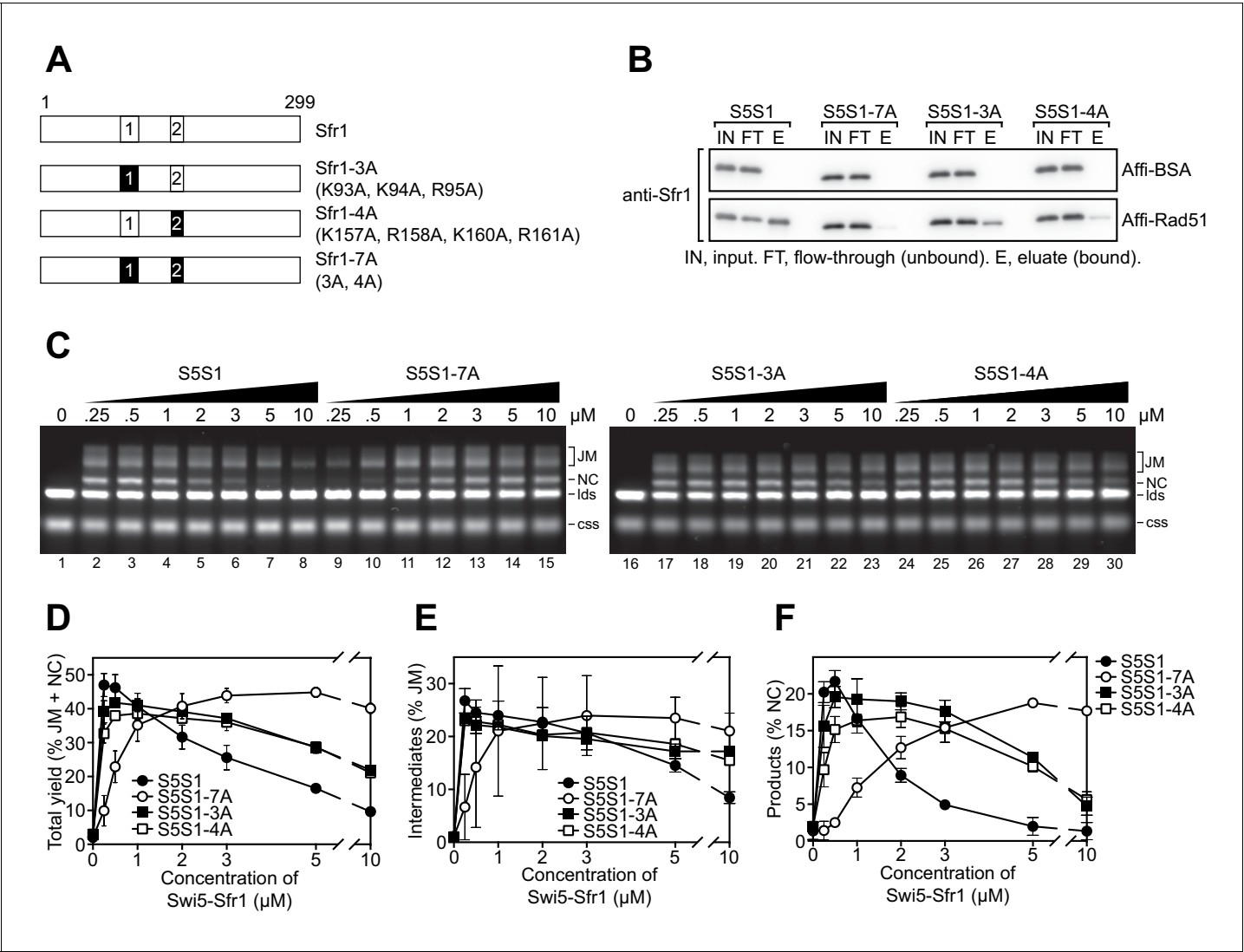

**Figure 5.** Interaction of sites 1 and 2 with Rad51 is important for the stimulation of Rad51-driven strand exchange. (A) Schematic of Sfr1-Rad51 interaction site mutants. (B) The interaction of Swi5-Sfr1 (S5S1, wild type or mutants) with Rad51 was investigated by a pull-down assay. Affi-BSA is a control for nonspecific binding. (C) Three-strand exchange reactions were performed with the indicated concentrations of Swi5-Sfr1 (wild type or mutants). The percentage of DNA signal per lane corresponding to total yield (D), JM (E) or NC (F) was plotted against Swi5-Sfr1 concentration. For (D–F), mean values of three independent experiments ± s.d. are shown.

The online version of this article includes the following source data and figure supplement(s) for figure 5:

Source data 1. Yield (%) of total DNA for data in *Figure 5D–F*.
Figure supplement 1. Identification of residues within sites 1 and 2 that are important for the functional interaction with Rad51.

Rad51 was crosslinked to Affi-gel matrix and mixed with Swi5-Sfr1. A substantial fraction of wild-type Swi5-Sfr1 was recovered in the eluate (Rad51-bound fraction), although some of the protein remained in the flow-through (unbound fraction; *Figure 5B*). By contrast, the amount of 3A and 4A mutant proteins detected in the eluate was reduced, with much of the protein remaining in the flow-through. The 7A mutant protein was barely detected in the eluate, indicating that the binding seen in the 3A mutant was dependent on Site 2 and the binding seen in the 4A mutant was dependent on Site 1. Comparable trends were observed in a co-IP assay that did not involve crosslinking of Rad51 (*Figure 5—figure supplement 1C*). Taken together, these results suggest that Sites 1 and 2 facilitate the binding of Swi5-Sfr1 to Rad51 in a cooperative manner.

In the 7A mutant, both Sites 1 and 2 are mutated but the remainder of the N-terminus is intact. Thus, the 7A mutant can be employed to test whether the N-terminal domain of Sfr1 has any significant role other than to facilitate binding to Rad51. We therefore proceeded to characterize the biochemical activities of the 7A mutant. The 3A and 4A mutants were included to glean further insights into the nature of Rad51 stimulation by Swi5-Sfr1.

While Swi5-Sfr1C can stimulate Rad51 activity despite the absence of Sfr1N, 5-to-10-fold more of the complex is required to achieve the same level of stimulation as full-length Swi5-Sfr1 (*Figure 1B* lanes 3 and 12; *Kuwabara et al., 2012*), suggesting that the interaction between Sfr1N and Rad51 is important for efficient stimulation of strand exchange. Consistent with the observed binding defect, substoichiometric concentrations of the 7A mutant failed to efficiently stimulate Rad51-driven strand exchange, with a higher concentration of mutant protein required to achieve a wild-type level of joint molecules (JM, reaction intermediates) and nicked-circles (NCs, reaction products; *Figure 5C*). At 0.25 µM, the defect of the 7A protein was more pronounced for NC (~15 fold reduction) than JM (~5 fold reduction), suggesting that the ability of Swi5-Sfr1 to stimulate both the initial pairing of homologous DNA and the subsequent strand transfer by Rad51 are defective when the interaction with Sites 1 and 2 is ablated (*Figure 5D–F*). By contrast, the 3A and 4A mutants were able to promote efficient JM and NC formation at substoichiometric concentrations. Nevertheless, the loss of Rad51 stimulation observed at higher concentrations of wild-type Swi5-Sfr1 was attenuated in the 3A and 4A mutants (*Figure 5C* lanes 6, 21 and 28), suggesting that this loss of stimulation occurs due to unproductive interactions with Rad51 or sequestration of DNA substrates by Swi5-Sfr1 (*Figure 6—figure supplement 1* and see Discussion). Consistent with this notion, efficient stimulation of Rad51 was maintained at higher concentrations of the 7A mutant (*Figure 5C* lanes 8 and 15) and Swi5-Sfr1C (*Figure 1B,C*; *Kuwabara et al., 2012*). Collectively, these results indicate that interactions between Rad51 and both Sites 1 and 2 are important for efficient stimulation of strand exchange.

## Rad51 filament stabilization and ATPase stimulation is mediated by Sites 1 and 2

To determine why stimulation of Rad51-driven strand exchange is inefficient when Sites 1 and 2 are mutated, the molecular roles of Swi5-Sfr1 were considered. At substoichiometric concentrations, Swi5-Sfr1 effectively stabilizes Rad51 presynaptic filaments (*Kurokawa et al., 2008*). Thus, it seemed feasible that the observed impairment in strand exchange might be explained by defects in Rad51 filament stabilization. To test this possibility, filament stability was examined by fluorescence anisotropy. When Rad51 binds to a fluorescently-labeled oligonucleotide and forms a filament, fluorescence anisotropy increases due to a retardation in the motion of the labeled oligonucleotide (*Figure 6A*). The dissociation of Rad51 is accompanied by a reduction in anisotropy, with the rate of decline reflective of Rad51 filament stability. Rad51-ssDNA filaments were formed in the presence of ATP and filament collapse was induced via dilution into reaction buffer containing ATP but lacking DNA and protein. In the absence of Swi5-Sfr1, the decrement in anisotropy was sharp and reached a value that was observed in the absence of protein (~0.1) within ~500 s. Inclusion of wild-type Swi5-Sfr1 resulted in a slower reduction in anisotropy, indicating that Rad51 filaments had been stabilized (*Figure 6B*). Strikingly, inclusion of the 7A mutant did not result in any obvious filament stabilization (*Figure 6C*). Furthermore, although both 3A and 4A mutants showed some stabilization of Rad51 filaments, the magnitude of this stabilization was less than that observed for wild-type protein (*Figure 6D,E*). Consistent with these observations, the reaction rate constants for dissociation of Rad51-ssDNA complexes ($k_{off}$) showed a substantial decline in the presence of Swi5-Sfr1, a lesser decline for the 3A and 4A mutants, and only a marginal decline for the 7A mutant (*Figure 6F*). Taken

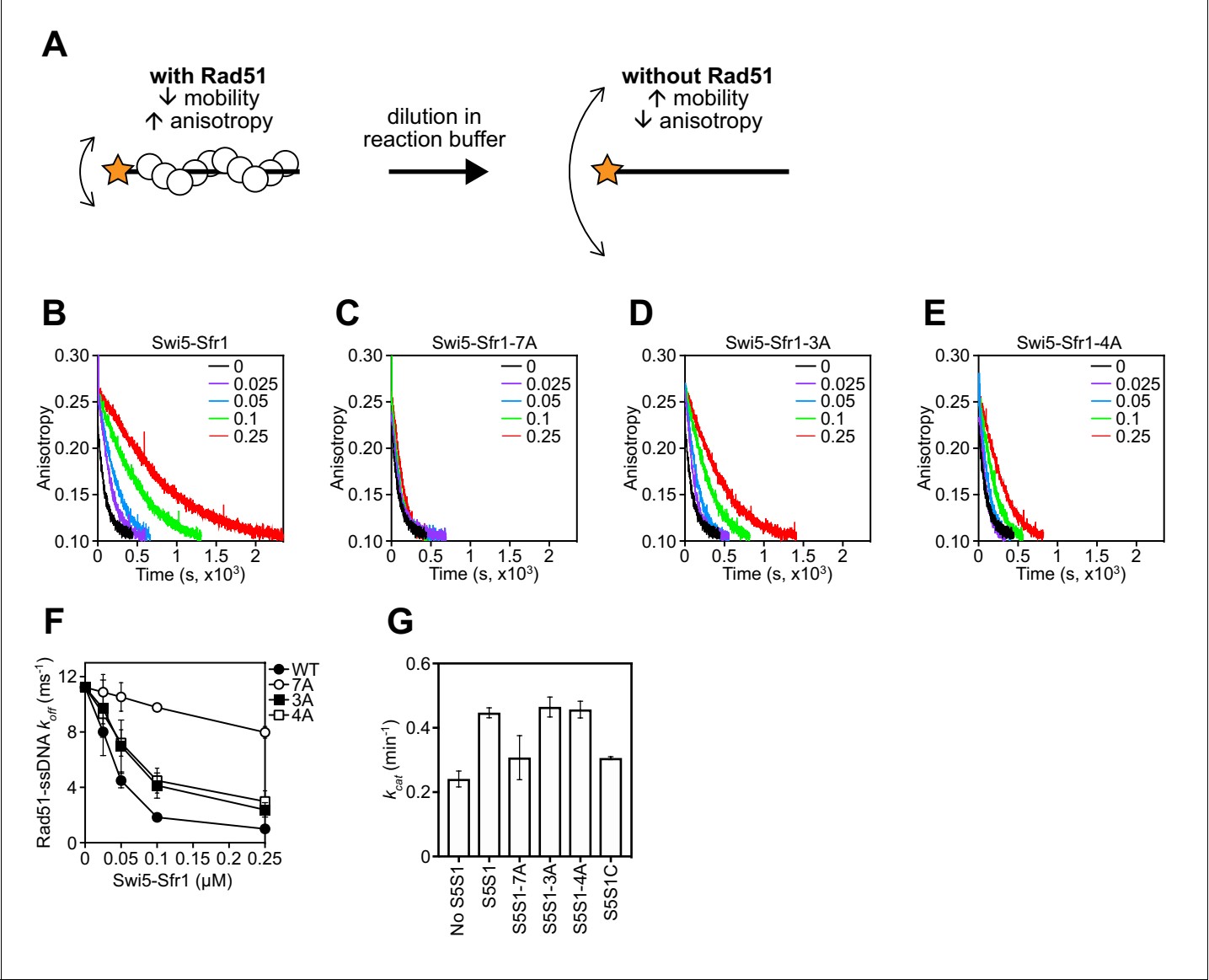

**Figure 6.** Rad51 filament stabilization and ATPase stimulation requires interactions with sites 1 and 2. (**A**) Schematic of the fluorescence anisotropy assay. Rad51 monomers (white circles) are depicted forming a filament on an oligonucleotide (black line) labeled with the TAMRA fluorophore (orange star). (**B–E**) The anisotropy of fluorescently labeled ssDNA in complex with Rad51 was monitored following induction of filament collapse in the presence of Swi5-Sfr1 (wild type or mutants) at the indicated concentrations (μM). (**F**) $k_{off}$ values were determined for Rad51-ssDNA complexes at the indicated concentrations of Swi5-Sfr1. (**G**) Rad51-dependent ATP turnover was determined in the presence of the indicated Swi5-Sfr1 variant. For (**F,G**), mean values of three independent experiments ± s.d. are shown.

The online version of this article includes the following source data and figure supplement(s) for figure 6:

**Source data 1.** Anisotropy values for data in *Figure 6B–E*.

**Figure supplement 1.** DNA binding is differentially defective in Site 1 and Site 2 mutants.

**Figure supplement 1—source data 1.** Unbound DNA (%) for data in *Figure 6—figure supplement 1B*.

together, these results indicate that Sites 1 and 2 within Sfr1N interact cooperatively with Rad51 to facilitate filament stabilization by Swi5-Sfr1.

In addition to stabilizing Rad51 filaments, Swi5-Sfr1 has been shown to stimulate the ATPase activity of Rad51, which is also important for efficient strand exchange (*Haruta et al., 2006*; *Kurokawa et al., 2008*; *Ito et al., 2018*). Since substoichiometric concentrations of Swi5-Sfr1C failed to efficiently stimulate the ATPase activity of Rad51 (*Kuwabara et al., 2012*), we sought to

determine whether Rad51-dependent ATP hydrolysis was potentiated by the 7A mutant. As expected, wild-type Swi5-Sfr1 was able to efficiently enhance ATP hydrolysis by Rad51 at substoichiometric concentrations (Swi5-Sfr1:Rad51 ratio of 1:20), with a 1.85-fold increase in ATP turnover (*Figure 6G*). By contrast, the 7A mutant only managed to enhance the ATPase activity of Rad51 1.28-fold, which is similar to the 1.27-fold stimulation observed with Swi5-Sfr1C. The 3A and 4A mutants stimulated ATP hydrolysis like wild type. These results suggest that interaction of either Site 1 or 2 with Rad51 is sufficient to promote efficient stimulation of ATP hydrolysis.

## DNA repair in Sfr1-Rad51 interaction mutants is dependent on Rad51 paralogs

To analyze the in vivo DNA repair activity of the Sfr1-Rad51 interaction mutants, strains were constructed in which the native *sfr1⁺* gene was replaced with either *sfr1-7A*, *sfr1-3A* or *sfr1-4A*. Unexpectedly, the interaction mutants did not show any obvious sensitivity to DNA damage (*Figure 7A*, *Figure 7—figure supplement 1A*), which is in sharp contrast to the DNA repair defect of the N-terminal deletion strain (*Figure 1D*, *Figure 1—figure supplement 1A,C*). A marginal sensitivity was observed for the *sfr1-7A* strain but this was not statistically significant (*Figure 7B*).

Previous genetic studies suggested that there are two HR sub-pathways in *S. pombe*: one dependent on Swi5-Sfr1 and the other dependent on the Rad51 paralogs Rad55-Rad57 (*Akamatsu et al., 2003*; *Akamatsu et al., 2007*). Thus, in the absence of Rad55/Rad57, it is possible to evaluate the Rad51-dependent DNA repair that is mediated solely by Swi5-Sfr1. For this purpose, the interaction mutants were introduced into the *rad55Δ* background. Strikingly, in the absence of Rad55, the *sfr1-7A* mutant showed the same DNA damage sensitivity as the *sfr1Δ* mutant (*Figure 7C*). Furthermore, both the *sfr1-3A* and *sfr1-4A* mutants were more sensitive to DNA damage than *sfr1⁺* in the *rad55Δ* background, although this sensitivity was not as severe as that observed for the *rad55Δ sfr1Δ* and *rad55Δ sfr1-7A* strains (*Figure 7C,D*). Similar results were obtained in the *rad57Δ* background (*Figure 7—figure supplement 1B*). These results indicate that the residues mutated in *sfr1-7A* are indeed important for DNA repair, while also suggesting that the DNA repair defects of Sfr1 amino acid-substitution mutants are suppressed by a Rad55-Rad57-dependent mechanism.

## Swi5-Sfr1 and Rad55-Rad57 form a Rad51-independent complex

One possible explanation for the above results is that Rad55-Rad57 plays a role in facilitating the recruitment of Swi5-Sfr1 to Rad51, perhaps by acting as a molecular bridge. A prerequisite of this model is that Rad55-Rad57 can bind to both Rad51 and Swi5-Sfr1. While an interaction between Rad57 and Rad51 has been reported by yeast two-hybrid analysis (*Tsutsui et al., 2001*), a possible interaction between Rad55-Rad57 and Swi5-Sfr1 was never examined due to the genetic evidence suggesting that they comprise independent sub-pathways of HR (*Akamatsu et al., 2003*; *Akamatsu et al., 2007*). To test if Rad55-Rad57 and Swi5-Sfr1 interact with each other, we sought to partially purify Rad55-Rad57. Protein A-tagged Rad55 and untagged Rad57 were co-expressed in *S. pombe* (*Figure 7—figure supplement 1C*). As a negative control, the same experiment was conducted in parallel using a strain transformed with empty vectors. Cleared cell lysates were incubated with IgG-agarose resin to enrich Protein A-tagged proteins. Rad55-Rad57 was then specifically eluted via the addition of 3C protease, which cleaves between Rad55 and resin-bound Protein A. Rad55 and Rad57 constituted the two major bands in this partially purified protein preparation and were present in seemingly stoichiometric amounts (*Figure 7—figure supplement 1D*), consistent with previous reports in budding yeast (*Sung, 1997a*; *Liu et al., 2011*).

Partially purified Rad55-Rad57 was incubated with purified Swi5-Sfr1 (wild type or 7A) and Rad57 was immunoprecipitated. The contents of these immunoprecipitates were then examined by immunoblotting. As expected, Rad55 was found to co-IP with Rad57 (*Figure 7E*). Moreover, both Swi5-Sfr1 and Swi5-Sfr1-7A coIP'd with Rad57. While partially purified Rad55-Rad57 appeared to be reasonably pure (*Figure 7—figure supplement 1D*), it remained possible that Rad51 had co-purified with Rad55-Rad57 throughout the purification process. If so, Rad51 bound to Rad55-Rad57 could interact with Swi5-Sfr1, leading to indirect co-IP of Sfr1 with Rad57. To test this possibility, the anti-Rad57 immunoprecipitates were probed with an anti-Rad51 antibody. Importantly, Rad51 was completely undetectable in these immunoprecipitates (*Figure 7F*), suggesting that Swi5-Sfr1 and Rad55-Rad57 directly interact in a Rad51-independent manner.

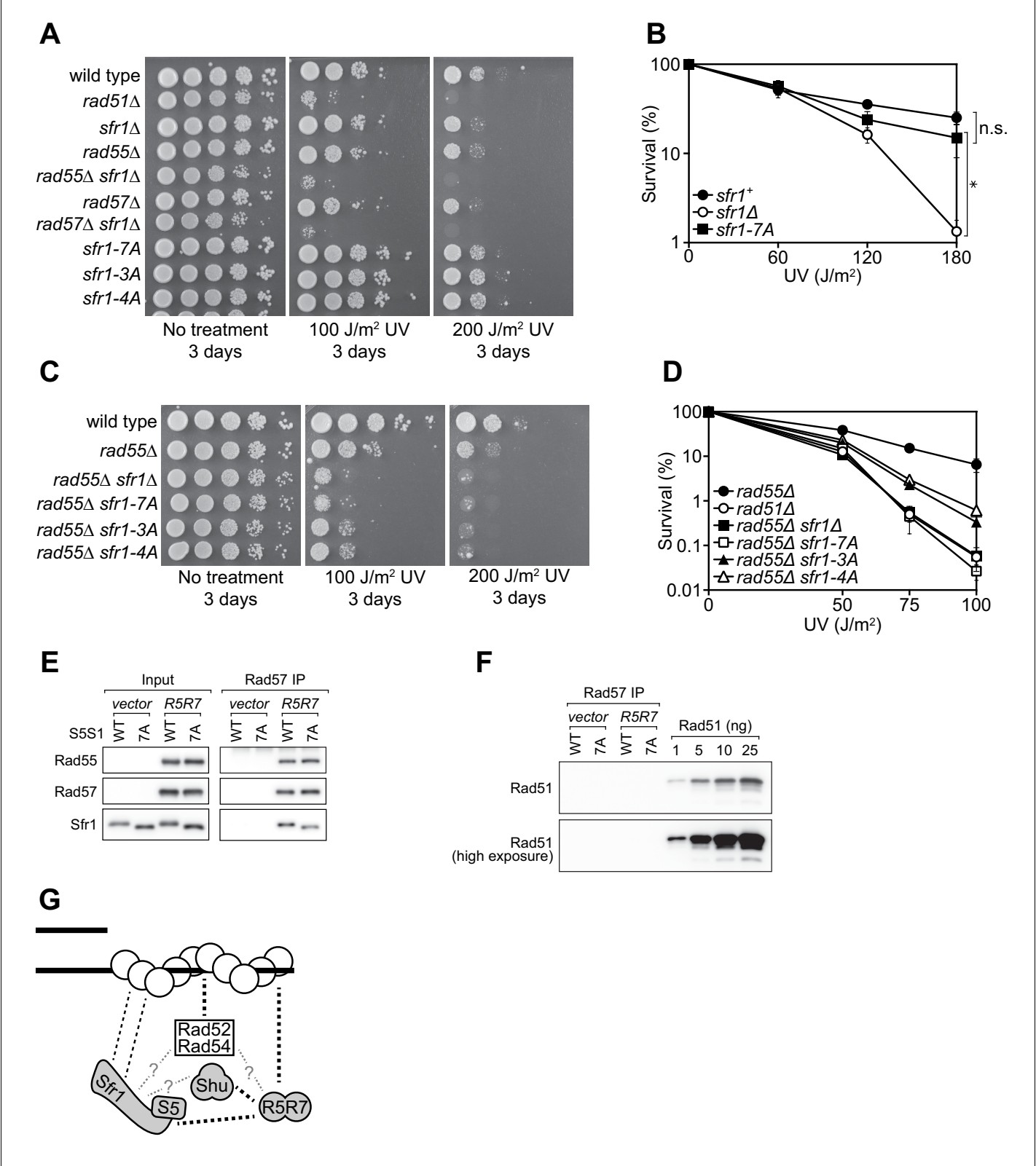

**Figure 7.** Rad55-Rad57 facilitates Swi5-Sfr1-dependent DNA repair. (A,C) 10-fold serial dilutions of the indicated strains were spotted onto rich media with or without acute UV treatment and grown at 30 ℃. (B,D) Percentage of cell survival was determined after acute exposure to UV. (E) The soluble cell lysate from *S. pombe* strains carrying the REP1 and REP2 expression vectors without inserts (*vector*) or encoding Rad55-Protein A and Rad57 (*R5R7*) was incubated with IgG-agarose. Proteins were eluted with 3C protease, for which a cleavage site exists between Rad55 and Protein A. This eluate was

*Figure 7 continued on next page*

*Figure 7 continued*

mixed with purified Swi5-Sfr1 (S5S1, wild type or 7A; input) and subjected to IP with anti-Rad57 antibody. These IPs were probed with the indicated antibodies. (F) Rad57 IPs from (E) and the indicated amounts of purified Rad51 were probed with an anti-Rad51 antibody. (G) Molecular bridging model. Rad51 monomers (white circles) are shown as a filament on ssDNA (solid black lines). Dashed black lines represent known physical interactions. Dashed gray lines with question marks represent potential physical interactions. Swi5-Sfr1 is recruited to Rad51 via two redundant mechanisms, one requiring its direct interaction with Rad51 and the other requiring Rad55-Rad57. See Discussion for details. For (B,D), mean values of three independent experiments ± s.d. are shown. Statistical analysis in (B) was by one-way ANOVA followed by Tukey's multiple comparisons test. *, p<0.05. n.s., not significant (p>0.05).

The online version of this article includes the following source data and figure supplement(s) for figure 7:

**Source data 1.** Survival (%) following UV irradiation for data in *Figure 7B* and *Figure 7D*.
**Figure supplement 1.** DNA repair in Sfr1-Rad51 interaction mutants is dependent on Rad57.

## Discussion

In this study, we characterized the interaction between Rad51, the key protein in HR, and Swi5-Sfr1, a widely conserved recombination auxiliary factor. The N-terminal half of Sfr1 was found to be essential for the role of Swi5-Sfr1 in promoting Rad51-dependent DNA repair (*Figure 1*). This domain was shown to be intrinsically disordered (*Figure 2*) and contain two sites that interact with Rad51 (*Figures 3* and *4*). Although mutation of the two interaction sites disrupted the physical and functional interaction with Rad51 in vitro (*Figures 5* and *6*), unexpectedly, defects in DNA repair were only observed in the absence of Rad55-Rad57, another auxiliary factor complex with which Swi5-Sfr1 was found to interact (*Figure 7*).

### Cooperative interactions with Rad51 are the primary function of Sfr1N

Although Sfr1N is not essential for stimulation of Rad51-driven DNA strand exchange (*Figure 1B,C*; *Kuwabara et al., 2012*), it is essential for the promotion of Rad51-dependent DNA repair by Swi5-Sfr1 (*Figure 1D*, *Figure 1—figure supplement 1A,C*). The fact that Swi5-Sfr1C can exert a stimulatory effect on Rad51 in vitro points towards the existence of a physical interaction between the two. However, the inability to detect such a complex, combined with the relative inefficiency with which Swi5-Sfr1C potentiates Rad51, strongly suggests that the interaction is too weak to observe by conventional means. This binding is likely augmented by the N-terminal fragment of Sfr1, which we posit functions as an anchor to keep Swi5-Sfr1C in close proximity to Rad51. Consistent with such a role for Sfr1N, NMR interaction analysis revealed that two domains within Sfr1N interact with Rad51 (*Figure 3*, *Figure 3—figure supplement 1*). Mutation of Site 1 or 2 weakened the interaction with Rad51, while mutation of both sites resulted in a near-complete loss of interaction (*Figure 5B*, *Figure 5—figure supplement 1C*), indicating that two sites within Sfr1N bind cooperatively to Rad51. Interestingly, Rad51-dependent strand exchange and ATP hydrolysis were significantly impaired only when both sites were mutated (*Figures 5C–F* and *6G*). These results indicate that the reduced interaction in the single site mutants is sufficient for Swi5-Sfr1 to fully stimulate Rad51 in these assays. Although this points toward functional redundancy between Sites 1 and 2, it is possible that these assays are not sensitive enough to detect marginal defects in the stimulation of Rad51. Indeed, the fluorescence anisotropy assay revealed a severe defect in Rad51 filament stabilization for the 7A mutant and a modest defect for the 3A and 4A mutants (*Figure 6B–F*), arguing that interaction of both Sites 1 and 2 with Rad51 is important for efficient filament stabilization. Swi5-Sfr1 has also been shown to stabilize Rad51 filaments against the F-box helicase Fbh1 (*Tsutsui et al., 2014*). It would be interesting to test whether the interaction mutants can function in a similar capacity.

The reduction in NMR signals from Sites 1 and 2 in the presence of Rad51 (*Figure 3E*), combined with the gradual chemical shift changes observed for some residues (*Figure 3C*, *Figure 3—figure supplement 1D*), indicated that the association and dissociation of Sfr1N and Rad51 is fast on the NMR timescale, suggesting that the Sfr1N-Rad51 interaction is relatively weak. While monomeric Rad51 (~40 kDa) is too small to cause severe line-broadening of NMR signals, the interaction of multimeric Rad51 (*Figure 3—figure supplement 2*) with Sfr1N could explain the drastic reduction in NMR signals for Sites 1 and 2.

These results were largely substantiated by site-specific crosslinking of residues within Site 1, and to a much lesser extent Site 2, to Rad51 (*Figure 4*, *Figure 4—figure supplement 1B*). Since both

sites are involved in electrostatic interactions with Rad51, the more robust crosslinking of Site 1 may be due to the added contribution of hydrophobic interactions between Site 1 and Rad51. One interpretation of these results is that the Site 1-Rad51 interaction is of higher affinity than the Site 2-Rad51 interaction. However, the results of co-IP experiments suggested that mutating Site 2 disrupts the Rad51 interaction to a greater extent than mutating Site 1 (compare 3A and 4A in *Figure 5B*, *Figure 5—figure supplement 1C*). When making such comparisons, it is important to note that the NMR and crosslinking experiments only involved Sfr1N, whereas the co-IP analysis involved full-length Sfr1 in complex with Swi5. It is possible that Site 2 plays a more prominent role in the interaction with Rad51 in the context of the full-length complex. The reduced crosslinking of Site 2 residues could also be explained by the fact that replacement of a given residue with the aromatic *p*BPA could itself disrupt the interaction with Rad51, especially since Site 2 is less hydrophobic than Site 1.

Previous structural analysis indicated that the self-association of Rad51 involves the conserved FxxA motif (where x can be any residue) and that the BRC3 and BRC4 repeats of BRCA2 bind to this motif through Rad51 mimicry, leading to destabilization of the Rad51 filament (*Shin et al., 2003*). This identified destabilization/stabilization of the interactions between Rad51 monomers as one way of regulating filament stability. A distinct method of stabilizing Rad51 filaments was uncovered for budding yeast Rad55-Rad57, which was shown to cap the end of Rad51 filaments and antagonize the filament destabilizing activity of the Srs2 anti-recombinase (*Liu et al., 2011*). Interestingly, neither Swi5 nor Sfr1 contains an FxxA motif. This is consistent with the identification of Rad51 interaction sites within the intrinsically disordered N-terminal domain of Sfr1 (*Figure 2*, *Figure 2—figure supplement 1*), which has undergone substantial sequence divergence (*Figure 5—figure supplement 1A*). We therefore favor the previously proposed model whereby Swi5-Sfr1C inserts into the wide grooves of the Rad51 nucleoprotein filament, locking the filament in an active conformation, with Sfr1N plastering along the outside of the filament to maintain Swi5-Sfr1C within the groove (*Fornander et al., 2014*; *Kokabu et al., 2011*; *Kuwabara et al., 2012*).

While this model could explain how Swi5-Sfr1 stabilizes the filament, it does not explain how Swi5-Sfr1 stimulates ATP hydrolysis and extensive strand transfer by Rad51, which are highly dynamic processes thought to involve dissociation of Rad51 from DNA (*Ito et al., 2018*). It is possible that the flexibility of Sfr1N allows Swi5-Sfr1 to remain bound to Rad51 despite conformational changes in the filament. This could also entail release of dissociating Rad51 molecules and re-binding of Sfr1N to molecules incorporated in the filament, thus preventing diffusion of Swi5-Sfr1 from the Rad51 filament. NMR interaction analysis suggested that binding of Sfr1N to Rad51 is relatively short-lived (see above), consistent with a model in which the dynamic association and dissociation of Swi5-Sfr1 from Rad51 plays a role in stimulation of DNA strand exchange. Sites 1 and 2 may comprise a single Rad51 interacting unit in 3D space, although they are unlikely to share the same interface on Rad51, given that Site 1 is more hydrophobic than Site 2. Alternatively, Site 1 may bind to Rad51 with Site 2 acting as a scaffold, consistent with the results of our crosslinking (*Figure 4C*, *Figure 4—figure supplement 1B*) and co-IP (*Figure 5B*, *Figure 5—figure supplement 1C*) experiments.

We note that the 7A mutant complex displayed a near-loss of DNA binding and the 4A mutant showed a clear defect in DNA binding (*Figure 6—figure supplement 1A*). A mild impairment in DNA binding became apparent upon closer inspection of the 3A mutant (*Figure 6—figure supplement 1B*). It is likely that these defects were caused by mutation of the positively charged Lys/Arg residues to Ala residues, leading to neutralization of the electrostatic attraction to DNA. Despite showing different levels of DNA binding, 3A and 4A were indistinguishable in all other aspects (*Figures 5–7*). Although it is formally possible that our observations with the 7A mutant are related to a defect in DNA binding, the high similarity of the results obtained with 3A and 4A argues that DNA binding by Swi5-Sfr1 is impertinent to its role in stimulating Rad51 activity or promoting Rad51-dependent DNA repair. Notably, mouse Swi5-Sfr1 (mSwi5-Sfr1) stimulates Rad51 through very similar mechanisms despite being unable to bind DNA (*Tsai et al., 2012*). This is in stark contrast to other recombination auxiliary factors such as Rad52, Rad54 and Hop2-Mnd1, whose ability to bind DNA is integral for recombinase stimulation (*Seong et al., 2008*; *Wright and Heyer, 2014*; *Zhao et al., 2014*). In addition to its involvement in DNA repair, human Sfr1 (hSfr1) has been implicated in transcriptional regulation (*Feng et al., 2013*; *Yuan and Chen, 2011*), raising the possibility that *S. pombe* Sfr1 may also have functions unrelated to DNA repair. In agreement with this, *Cipak et al. (2009)* reported that Swi5-Sfr1 forms a complex with the XPG-family RNA nuclease

Mkt1 (SPAC139.01c), which was recently shown to be involved in RNAi-mediated silencing and establishment of heterochromatin (*Taglini et al., 2019*). We speculate that the ability of Sfr1 to bind DNA may be related to an as yet uncharacterized function.

## Rad51 paralogs promote Swi5-Sfr1-dependent DNA repair

While the *sfr1Δ* and *rad55Δ* single mutants are sensitive to DNA damage, neither is as sensitive as *rad51Δ*, which is epistatic to both (*Akamatsu et al., 2003*; *Khasanov et al., 1999*). However, because the *sfr1 rad55* double mutant shows the same sensitivity as *rad51Δ* (e.g., *Figure 7A*), it was concluded that two independent sub-pathways of HR exist in *S. pombe* (*Akamatsu et al., 2003*). Despite the numerous defects observed in vitro, the *sfr1-7A* mutant strain was proficient for DNA repair, but this repair was dependent on Rad55-Rad57 (*Figure 7A–D*), indicating that a Rad55-Rad57-dependent mechanism overcomes defects in the binding of Swi5-Sfr1 to Rad51.

To explain these results, we propose that the interaction of Swi5-Sfr1 with Rad51 is enabled by two redundant mechanisms: one through a direct interaction involving Sites 1 and 2 in Sfr1N and the other through Rad55-Rad57, which interacts with Rad51 (*Tsutsui et al., 2001*) and acts as a molecular bridge to facilitate the recruitment of Swi5-Sfr1 to Rad51. Hence, although Swi5-Sfr1-7A cannot interact directly with Rad51, Rad55-Rad57 aids the recruitment of Swi5-Sfr1-7A to Rad51, allowing it to exert a stimulatory effect on Rad51; this would explain why *sfr1-7A* is proficient for DNA repair. However, in the absence of Rad55/Rad57, this tethering is lost but Swi5-Sfr1 can nevertheless promote some DNA repair via its direct interaction with Rad51, thus explaining why *rad55Δ* is not as sensitive as *rad51Δ*. It is only when both interaction mechanisms are defective, as in the *rad55Δ sfr1-7A* strain, that the promotion of Rad51-mediated DNA repair by Swi5-Sfr1 is completely lost. Consistent with this possibility, Swi5-Sfr1 was found to form a Rad51-independent complex with partially purified Rad55-Rad57 (*Figure 7E,F*). Critically, Swi5-Sfr1-7A was also able to form a complex with Rad55-Rad57. The apparent absence of Rad51 (*Figure 7F*)—the major interacting partner of Rad55-Rad57—combined with the reasonably high purity achieved by the one-step purification process (*Figure 7—figure supplement 1D*) implies that this complex formation involves direct binding of Swi5-Sfr1 to Rad55-Rad57, in support of the proposed model. We therefore surmise that, while the Swi5-Sfr1 and Rad55-Rad57 sub-pathways are capable of operating independently of each other, as observed in the *rad55Δ* and *sfr1Δ* backgrounds, Swi5-Sfr1 and Rad55-Rad57 likely collaborate to promote Rad51-dependent DNA repair in wild-type cells.

Rad55-Rad57 facilitates recruitment of the Shu complex to Rad51 by binding to both and acting as a molecular bridge (*Gaines et al., 2015*; *Khasanov et al., 2004*), so it could plausibly fulfil a similar role for Swi5-Sfr1. However, unlike the Shu complex, Swi5-Sfr1 can interact directly with Rad51, so any contribution made by Rad55-Rad57 to this interaction would enhance rather than enable complex formation with Rad51. The requirement for such a mechanism may stem from the fact that the direct interaction between Swi5-Sfr1 and Rad51 is relatively weak (see above). Indeed, previous attempts by us and others to co-IP Swi5-Sfr1 and Rad51 from yeast extracts has been unsuccessful (*Akamatsu et al., 2003*; *Cipak et al., 2009*), suggesting that the cellular interaction is too weak to capture. It is tempting to speculate that Rad55-Rad57, Swi5-Sfr1, and the Shu complex exist as a higher-order auxiliary factor complex, perhaps as part of a Rad52-containing DNA repair center (*Lisby et al., 2001*; *Lisby et al., 2003*). Evidence for the existence of such a complex, along with the delineation of the relationship between Swi5-Sfr1 and Rad55-Rad57, will be the focus of future research.

Interestingly, unlike *sfr1-7A*, *sfr1C* showed the same DNA damage sensitivity as *sfr1Δ* even in the presence of Rad55-Rad57 (*Figure 1D*, *Figure 1—figure supplement 1A,C*). This suggests that, in addition to its primary role in anchoring Swi5-Sfr1 to Rad51, Sfr1N may have a secondary role in coordinating the collaboration between Swi5-Sfr1 and Rad55-Rad57.

## Evolutionary conservation of Sfr1N structure and function

Although the role of Swi5-Sfr1 in promoting Rad51-dependent DNA repair is conserved in mammals (*Akamatsu and Jasin, 2010*; *Argunhan et al., 2017a*; *Lu et al., 2018*; *Su et al., 2014*; *Su et al., 2016*; *Yuan and Chen, 2011*), the precise mode of interaction with Rad51 appears to have undergone some divergence (*Tsai et al., 2012*; *Yuan and Chen, 2011*). In *Saccharomyces cerevisiae*, the Swi5-Sfr1 homolog Sae3-Mei5 is produced only during meiosis and functions exclusively in the Dmc1

branch of meiotic HR (*Hayase et al., 2004*; *Tsubouchi and Roeder, 2004*). Both Swi5-Sfr1 and Sae3-Mei5 stimulate the strand exchange activity of Dmc1 via similar mechanisms (*Ferrari et al., 2009*; *Haruta et al., 2006*; *Murayama et al., 2013*), and although Sae3-Mei5 does interact directly with Rad51, it does not stimulate the activity of Rad51 (*Cloud et al., 2012*; *Say et al., 2011*), unlike Swi5-Sfr1.

A consistent trend across all examined species is that Sfr1 plays some role in facilitating the interaction with the recombinase partner. Since the amino acid sequence of the N-terminal half of Sfr1 shows little conservation compared to the C-terminal half (*Figure 5—figure supplement 1A*), it is tempting to ascribe the similarities among species to the C-terminus. However, sequence divergence across large evolutionary distances does not necessarily reflect a lack of functional conservation for intrinsically disordered regions, which accumulate mutations at a higher rate than structured domains (*Brown et al., 2002*). Notably, the large subunit of RPA contains an intrinsically disordered region whose function is conserved despite significant sequence divergence (*Daughdrill et al., 2007*), raising the possibility that the structure and/or function of Sfr1N is conserved. While empirical evidence is lacking, disorder predictions (*Jones and Cozzetto, 2015*) for the N-terminal half of *S. pombe* Sfr1 agree with the data presented here (*Figure 2—figure supplement 2A*), and similar profiles were generated for Sfr1 from *Schizosaccharomyces japonicus* and *Schizosaccharomyces octosporus* (*Figure 2—figure supplement 2B,C*). Furthermore, the N-terminal halves of mSfr1, hSfr1 and Mei5 are predicted to be enriched in intrinsically disordered regions (*Figure 2—figure supplement 2D–F*). This analysis also highlighted potential protein binding sites within the N-terminal halves of *S. japonicus* Sfr1, hSfr1 and Mei5. In support of this, the N-terminal half of Mei5 has already been shown to interact with Dmc1 (*Hayase et al., 2004*; *Say et al., 2011*). Thus, in addition to the conserved function of Swi5-Sfr1 in promoting HR, the intrinsically disordered nature of Sfr1's N-terminus and its role in facilitating interactions with recombinases may be evolutionarily conserved. Further studies will be required to test the validity of this prediction.

For the Key Resources Table, please see *Supplementary file 1*.

## Contact for reagent and resource sharing

Further information or requests for resources should be directed to and will be fulfilled by the Lead Contact, Hiroshi Iwasaki (hiwasaki@bio.titech.ac.jp).

## Materials and methods

### *S. pombe* strains

All strains are listed in the Key Resources Table (*Supplementary file 1*). Except for BA1 (*Msmt-0 leu-1–32 ura4-D18 arg3-D1 isp6::hphMX4 psp3::kanMX4*), all *S. pombe* strains are isogenic derivatives of strain YA119 (*Akamatsu et al., 2003*); *Msmt-0 leu-1–32 ura4-D18 his3-D1 arg3-D1*). Standard media was used for growth (YES), selection (YES with drugs or EMM), and sporulation (SPA), as described previously (*Hentges et al., 2005*).

### *E. coli* strains

All strains are listed in the Key Resources Table (*Supplementary file 1*). *E. coli* strains were constructed by transforming BL21 (DE3) containing the pEVOL-pBpF plasmid (*Young et al., 2010*) with a pBKN220 plasmid (*Haruta et al., 2006*) encoding Sfr1N (with or without a TAG mutation) and a pET28a plasmid encoding Rad51. Thus, strains only differ in the expression of Sfr1N or Rad51 and only these differences are indicated in the Key Resources Table (*Supplementary file 1*). A '−" sign indicates that cells were transformed with an empty vector, whereas a '+" sign signals the presence of Sfr1N or Rad51 on that vector. If a codon in Sf1N was mutated to TAG, the mutated residue is listed instead of a '+" sign. Strains are listed in order of appearance. Standard media was used for growth (LB) and selection (LB with antibiotics), unless otherwise indicated.

### DNA damage sensitivity

A single colony was resuspended in 2 mL of YES and grown for 24 hr (*rad*[+]) or 48 hr (*rad*[−]). Cells from these cultures were then seeded into 2 mL of fresh YES and grown for ~14 hr until they reached log phase (~0.8×10$^7$ cells/mL). Cell density was adjusted to 2 × 10$^7$ cells/mL, 10-fold serial dilutions

were made, and 5 µL of each dilution was spotted onto YES plates (no treatment control) or YES plates containing the indicated genotoxins. For UV irradiation, cells were spotted onto YES plates and treated with acute UV exposure of the indicated dose. The leftmost spot on each plate contains $1 \times 10^5$ cells. Cells were photographed with a digital camera after growth at 30 °C (2–4 days, as indicated). For clonogenic assays, cells were grown as described above and spread onto several YES plates and irradiated with the indicated dose of UV. After 3 ($rad^+$) or 4 ($rad^-$) days of growth, colonies were counted.

## Extraction of cellular proteins for immunoblotting

Cells ($1 \times 10^8$) were harvested and processed exactly as previously described (**Argunhan et al., 2017b**). Briefly, harvested cells were resuspended in 1 mL of ice-cold water. 150 µL of lysing solution (1.85 M NaOH 7.5% beta-mercaptoethanol) was added and mixed with the cells, followed by a 15 min incubation on ice. 150 µL of 55% TCA was added, followed by a further 10 min incubation on ice. Precipitated proteins were pelleted by centrifugation (16,000 $g$ 10 min 4 °C) and dissolved with mixing (65 °C 10 min) in 100 µL of urea loading buffer (8 M urea, 5% SDS, 200 mM Tris-Cl pH 6.8, 1 mM EDTA, 0.01% BPB, freshly supplemented with 10% vol each of 1 M DTT and 2 M Tris). Proteins were separated by SDS-PAGE, transferred to PVDF membranes, and detected with the indicated antibodies.

## CD spectrometry

CD measurements were made using a Jasco J-720W spectrometer with a Peltier temperature controller. Sfr1N (4 µM) in buffer N (20 mM sodium phosphate [pH 6], 25 mM NaCl, 1 mM DTT) was placed in a 0.1 cm path length quartz cuvette. The CD spectrum was acquired from 180 nm to 260 nm at 25°C with a 1.0 nm bandwidth, 0.5 nm resolution, 50 nm/min scan speed, with 1 s averaging at each wavelength. Three spectra were averaged to give the spectrum of the protein and blank spectrum measured for buffer N alone was subtracted to produce the final spectrum.

## NMR analysis of Sfr1N

Sfr1N (residues 1–176) was subcloned into pBKN220 (**Haruta et al., 2006**), which was transformed into the *E. coli* strain BL21 (DE3) RIPL. Plasmids containing Sfr1N variants were prepared by using the protocol in the QuikChange Site-Directed Mutagenesis Kit (Agilent).

For the production of uniformly $^{15}$N-labeled or $^{13}$C and $^{15}$N-labeled proteins, *E. coli* cells were grown in M9 media supplemented with $^{15}$NH$_4$Cl (1 g/L, Cambridge Isotope Laboratories) or with $^{15}$NH$_4$Cl and $^{13}$C$_6$-glucose (3 g/L, SHOKO Science), respectively. For the production of amino acid selectively labeled proteins, modified M9 media supplemented with a $^{15}$N-labeled amino acid (either Ala [200 mg/L]), Arg [200 mg/L], Ile [100 mg/L], Leu [100 mg/L], Lys [100 mg/L] or Phe [50 mg/L], all from Cambridge Isotope Laboratories) and other non-labeled amino acids (400 mg/L Ala, 400 mg/L Arg, 250 mg/L Asp, 50 mg/L Cys, 400 mg/L Glu, 400 mg/L Gly, 100 mg/L His, 100 mg/L Ile, 100 mg/L Leu, 150 mg/L Lys, 50 mg/L Met, 50 mg/L Phe, 150 mg/L Pro, 1000 mg/L Ser, 100 mg/L Thr, 50 mg/L Trp, 100 mg/L Tyr, and 100 mg/L Val) was used for culturing cells. Cultures were shaken in baffled flasks at 37°C until the OD$_{600}$ reached 0.8 ~ 1.0. Protein expression was then induced by the addition of 0.5 mM isopropyl-β-D-thiogalactoside (IPTG) for 20 hr at 20°C. For cultures with labeled amino acids, the induction was limited to ≤4 hr to minimize isotope scrambling. Sfr1N was then purified as described (**Kuwabara et al., 2010**), except the storage buffer used was buffer N (20 mM sodium phosphate [pH 6], 25 mM NaCl, 1 mM DTT). A truncated version of Sfr1N (127-176) was purified as a fusion protein with an N-terminal maltose-binding protein tag. Following purification with amylose resin (NEB), the tag was cleaved with Factor Xa and the cleaved peptide was isolated by ultrafiltration (Amicon Ultra-15 MWCO 10K, Merck) and subsequent solid phase extraction using a Sep-Pak C8 plus short cartridge (Waters).

NMR experiments were carried out using Bruker Avance III HD 500 and 800 spectrometers equipped with TCI cryoprobes at 25°C. The spectra were processed using the program NMRPipe (**Delaglio et al., 1995**) and analyzed using the program Sparky (Goddard, T. D. and Kneller, D. G. University of California, San Francisco).

For the main-chain resonance assignments of Sfr1N, $^{13}$C and $^{15}$N doubly labeled samples at concentrations of 0.15 ~ 0.20 mM in buffer N mixed with 5% D$_2$O were prepared and placed in 5 mm

symmetrical microtubes (Shigemi). The $^1$H-$^{15}$N HSQC spectrum (*Kay et al., 1992*; *Grzesiek and Bax, 1993*) was acquired at a $^1$H frequency of 800 MHz with a scan number of 8, 1024 complex points and an acquisition time of 91.8 ms in the observed dimension, and 256 complex points and an acquisition time of 132 ms in the indirect dimension. The HNCA (*Grzesiek and Bax, 1992*; *Kay et al., 1994*), HN(CO)CA (*Grzesiek and Bax, 1992*; *Kay et al., 1994*), HNCACB (*Muhandiram and Kay, 1994*), HN(CO)CACB (*Yamazaki et al., 1994*), HNCO (*Grzesiek and Bax, 1992*; *Kay et al., 1994*), and HN(CA)CO (*Werner-Allen et al., 2006*) spectra were acquired at 800 MHz with a scan number of 8. All spectra were acquired with 512 complex points and an acquisition time of 45.9 m in the observed dimension. In the $^{15}$N dimension, all spectra except HNCACB were acquired with 45 complex points and an acquisition time of 23.1 ms, while the HNCACB spectrum was measured with 47 complex points and an acquisition time of 24.1 ms. The experiments with $^{13}$C$_\alpha$, $^{13}$C$_\alpha$/$^{13}$C$_\beta$, and $^{13}$CO evolution were acquired with 64, 128, and 43 complex points and acquisition times of 13.3, 10.6, and 8.90 ms in the $^{13}$C dimension, respectively.

To verify the signal assignments, the following samples were prepared and their $^1$H-$^{15}$N HSQC spectra were obtained at 500 MHz; amino-acid selectively $^{15}$N-labeled (Ala, Arg, Ile, Leu, Lys, or Phe) Sfr1N(1-176), Arg-selectively $^{15}$N-labeled R97A variant of Sfr1N(1-176), Lys-selectively $^{15}$N-labeled K93A variant of Sfr1N(1-176), and uniformly $^{15}$N-labeled Sfr1 (127-176). From the 160 expected main-chain amide NH signals, 157 were detected (98%) and assigned to specific residues in Sfr1N. The remaining three signals from Q2, S3, and H51 could not be assigned due to line-broadening. NMR data were deposited in the Biological Magnetic Resonance Bank (BMRB) repository with accession number 27885.

## Secondary structure analysis

The secondary structural elements were analyzed by calculating deviations of the observed $^{13}$C$_\alpha$, $^{13}$C$_\beta$, and $^{13}$CO chemical shifts from their residue-dependent random coil values (*Wishart and Sykes, 1994*; *Wishart et al., 1995*). Residues were deemed to form random coils if they displayed secondary chemical shift values within a limited range (between −0.7 and 0.7 for $^{13}$C$_\alpha$ and $^{13}$C$_\beta$ atoms, and between −0.5 and 0.5 for $^{13}$CO atoms of non-proline residues, and −4 to 4 for all three $^{13}$C atoms of proline residues). The program TALOS+ (*Shen et al., 2009*) was also used to predict the secondary structural units where $^1$H$_N$, $^{13}$C$_\alpha$, $^{13}$C$_\beta$, $^{13}$CO, and $^{15}$N$_H$ chemical shifts were used as input data. Predictions of disorder and protein binding sites for Sfr1 orthologs were generated by DISOPRED3 (*Jones and Cozzetto, 2015*).

## Heteronuclear NOE

The heteronuclear {$^1$H}-$^{15}$N NOE experiment (*Kay et al., 1989*; *Farrow et al., 1994*) was carried out for uniformly $^{15}$N-labeled Sfr1N at 800 MHz. The NOE values were determined from the ratio $I_{NOE}$/$I_{ref}$, where $I_{NOE}$ and $I_{ref}$ indicate the signal intensities in the spectra acquired with and without 3 s $^1$H presaturation, respectively. The NOE and reference spectra were acquired in an interleaved manner with a scan number of 32, 1024 complex points and an acquisition time of 91.8 ms in the observed dimension, and 256 complex points and an acquisition time of 132 ms in the indirect dimension.

## NMR interaction analysis

250 μL of uniformly $^{15}$N-labeled Sfr1N at a concentration of 0.1 mM in buffer N was mixed with a 0.1 mM Rad51 solution in buffer N at Sfr1N:Rad51 molar ratios of 1:0, 1:0.25, 1:0.5, 1:0.75, and 1:1. These Sfr1N-Rad51 mixtures were concentrated to 250 μl (Amicon Ultra-4 MWCO 10K, Merck) and used for NMR measurements. For each mixture, $^1$H-$^{15}$N HSQC spectra were acquired at 500 MHz with a scan number of 16, 1024 complex points and an acquisition time of 146 ms in the observed dimension, and 256 complex points and an acquisition time of 126 ms in the indirect dimension.

## Site-specific crosslinking

*E. coli* strains used in this study are listed in the Key Resources Table (*Supplementary file 1*). Experiments were performed essentially as described (*Miyazaki et al., 2016*). Co-expression of Sfr1N (residues 1–176, with or without a specific amber codon) and Rad51 was induced with 1 mM IPTG at an OD$_{600}$ of ~0.35 in *E. coli* strain BL21 (DE3) containing the pEVOL-pBpF plasmid either in the presence or absence of 1 mM *p*BPA. After 1 hr at 30 ˚C, cultures were pre-chilled on ice for 5 min before

250 µL of cells were spotted in a radial manner onto petri dishes at 4 °C and UV irradiated for 10 min at a distance of 4 cm with a B-100AP UV lamp (365 nm; UVP, LLC). 200 µL of cells was recovered from the plate and pelleted by centrifugation (20,000 $g$ 5 min 4 °C). This pellet was dissolved in 70 µL of urea loading buffer (*Argunhan et al., 2017b*) and subjected to SDS-PAGE followed by immunoblotting.

## Purification of proteins for biochemical analyses

Previously published protocols were followed to purify Rad51 (*Kurokawa et al., 2008*), RPA (*Haruta et al., 2006*), and Swi5-Sfr1 (wild type (*Haruta et al., 2006*), Sfr1N (*Kuwabara et al., 2010*), and Swi5-Sfr1C (*Kuwabara et al., 2010*). Swi5-Sfr1 mutants (3A, 4A, 7A) were purified by the same method as wild-type Swi5-Sfr1 except they were diluted to 25 mM NaCl instead of 100 mM NaCl before being applied to the HiTrap Heparin column.

For the partial purification of Rad55-Rad57, Rad55 with a C-terminal dual tag (7xHis-Protein A) was co-expressed with untagged Rad57 from the REP1 and REP2 plasmids, respectively, in a protease-deficient *S. pombe* strain (BA1). As a negative control, the same base strain was transformed with the REP1 and REP2 plasmids and treated exactly the same. After 20 hr in EMM without thymine, 400 mL of culture was harvested, washed with yeast wash solution (25 mM HEPES-KOH [pH 7.5], 150 mM NaCl, 1 mM PMSF) and stored as aliquots of $2 \times 10^9$ cells in −80 °C until required. An aliquot of cells was resuspended in 400 µL of yeast lysis buffer (YLB; 50 mM HEPES-KOH [pH 7.5], 500 mM NaCl, 5 mM Mg(OAc)$_2$, 5% glycerol, 0.05% igepal CA630, 1 mM ATP, 0.25 mM TCEP, 1 mM PMSF, 1x protease inhibitor cocktail [Roche]) and lysis was performed with ~500 µm glass beads using a Yasui Kikai Multi-Beads shocker (12 cycles, 30 s on, 30 s off, 2700 rpm). Following sequential clarification (20,000 g 10 min 2 °C, 20,000 $g$ 5 min 2 °C), the soluble cell extract was incubated with IgG-agarose resin (Sigma) for 3 hr at 4 °C. Resin-bound proteins were washed with YLB (500 µL x3) then buffer H200 (25 mM HEPES-KOH [7.5], 200 mM NaCl, 5 mM Mg(OAc)$_2$, 5% glycerol, 1 mM ATP, 0.25 mM TCEP; 500 µL x2). The resin was then resuspended in 200 µL of H200 and Rad55-Rad57 was eluted by incubating with PreScission protease (GE Healthcare), which cleaves between the 7xHis and Protein A tag on Rad55. 3.75 µL or 7.50 µL of this IgG-agarose eluate was mixed with an equal volume of 2x SDS loading buffer and subjected to purity analysis by SDS-PAGE and staining with CBB (*Figure 7—figure supplement 1D*).

All proteins were free of contaminating nuclease, protease and ATPase activity for the duration of the relevant assays. In all assays where comparisons were made between reactions with and without protein, the equivalent volume of protein storage buffer was added instead of the protein.

## Three-strand exchange assay

Strand exchange buffer (30 mM Tris-Cl [pH 7.5], 100 mM NaCl, 20 mM KCl, 3.5 mM MgCl$_2$, 2 mM ATP, 1 mM DTT, 5% glycerol, 8 mM phosphocreatine, 8 units/mL creatine phosphokinase) containing 10 µMnt PhiX174 virion DNA (NEB) was supplemented with 5 µM Rad51 and incubated at 37 °C for 10 min. The indicated concentration and variant of Swi5-Sfr1 was then added and the reaction was incubated for 10 min at 37 °C. Next, 1 µM of RPA was added and the reaction was incubated for 7 min at 37 °C. The reaction was initiated through the addition of 10 µMnt PhiX RF I DNA (NEB) linearized with ApaLI and incubated for a further 2 hr at 37°C. The 10 µL reactions were supplemented with 1 µL of psoralen (200 mg/mL) and subjected to psoralen-UV crosslinking to capture labile DNA structures. 1.95 µL of stop solution was then added (30 mM Tris-Cl [pH 7.5], 40 mM EDTA, 3% SDS, 5 mg/mL proteinase K). Following a 15 min incubation at 37°C, DNA was resolved in 1% agarose gels and stained with SYBR Gold (Thermo Fisher Scientific).

## In vitro interaction assay

For the Affi-gel interaction assay (*Figure 5B*), BSA or Rad51 was covalently attached to Affi-gel 15 (2 µg protein/µL gel) according to the manufacturer's instructions. 2 µg of the indicated Swi5-Sfr1 variant was diluted into 300 µL of Affi-gel buffer (25 mM HEPES-KOH [pH 7.5], 150 mM NaCl, 3.5 mM MgCl$_2$, 0.5 mM DTT, 0.05 µg/µL BSA, 10% glycerol, 0.05% igepal CA-630), input sample was taken, and 140 µL of this solution was mixed with 10 µL of Affi-BSA or Affi-Rad51. Reactions were then incubated with gentle mixing (30 °C 30 min). Following a brief centrifugation, flow-through samples were taken and the resin was washed with Affi-gel buffer (200 µL, x2). Bound proteins were eluted in

50 µL of SDS loading buffer with gentle mixing (37 ℃ 15 min). 9 µL (input, flow-through) or 3 µL (eluate) of sample was separated by SDS-PAGE, proteins were transferred to PVDF membranes and Sfr1 was detected with an anti-Sfr1 antibody (*Haruta et al., 2006*).

For the IP experiment in *Figure 5—figure supplement 1C*, 250 nM of a Swi5-Sfr1 variant and 250 nM of Rad51 were mixed on ice in 120 µL of IP buffer (30 mM Tris-Cl [pH 7.5], 150 mM NaCl, 3.5 mM MgCl$_2$, 5% glycerol, 0.1% Igepal CA-630). Input sample was taken and 100 µL of the solution was incubated at 30 ℃ for 15 min. Dynabeads Protein A (ThermoFisher) preincubated with anti-Rad51 antibody (*Haruta et al., 2006*) was added and mixtures were incubated with gentle mixing (4 ℃ 2 hr). Beads were washed with IP buffer (500 µL, x3) and bound proteins were eluted in 50 µL of SDS loading buffer with gentle mixing (65 ℃ 10 min). Proteins were separated by SDS-PAGE, transferred to PVDF membranes and detected with the indicated antibodies.

For the IP experiments in *Figures 7E,F* and 50 µL of the IgG-agarose eluate from a strain transformed with empty vectors or Rad55-Rad57 containing vectors was mixed on-ice with 45 µL of buffer H200 (25 mM HEPES-KOH [7.5], 200 mM NaCl, 5 mM Mg(OAc)$_2$, 5% glycerol, 1 mM ATP, 0.25 mM TCEP) and 5 µL of 20 µM Swi5-Sfr1 (wild type or 7A). 5 µL of input sample was taken, diluted with 20 µL of milliQ and mixed with an equal volume of 2x SDS loading buffer. The remaining 95 µL of reaction was incubated at 30 ℃ for 10 min then 4 ℃ for 5 min. Dynabeads Protein A (ThermoFisher) preincubated with anti-Rad57 antibody (*Tsutsui et al., 2001*) was added and mixtures were incubated with gentle mixing (4 ℃ 1 hr). Beads were washed with buffer H200 (300 µL, x3) and bound proteins were eluted in 50 µL of SDS loading buffer with gentle mixing (65 ℃ 10 min). Either 1 µL (Rad57), 3 µL (Rad55) or 5 µL (Sfr1 and Rad51) of eluate was separated by SDS-PAGE, transferred to PVDF membranes and detected with the indicated antibodies.

## Antibodies used for IP and immunoblotting

For the immunoblots of site-specific crosslinking experiments: anti-Rad51 (Rb 1:10,000; provided by Hiroshi Iwasaki); anti-Sfr1 (Rb 1:5,000; provided by Hiroshi Iwasaki). For the immunoblots of cellular proteins: anti-MYC (Rb 1:1,000; Sigma Aldrich C3956), anti-Rad51 (Rat 1:10,000; provided by Hiroshi Iwasaki), and anti-tubulin (Mu 1:10,000; Sigma Aldrich T5168). For the detection of Sfr1 in the Affi-gel assay: anti-Sfr1 (Rb 1:5,000; provided by Hiroshi Iwasaki). For IP of Rad51 complexes, anti-Rad51 (Rb; provided by Hiroshi Iwasaki) was used, and for detection by immunoblotting: anti-Rad51 (Rat 1:10,000; provided by Hiroshi Iwasaki), anti-Sfr1 (Mu 1–5 and 76–80, 1:1000 each; provided by Hiroshi Iwasaki), anti-Rad55 (1:5,000; provided by Hiroshi Iwasaki), and anti-Rad57 (1:5,000; provided by Hiroshi Iwasaki). For IP of Rad57 complexes, anti-Rad57 (Rb; provided by Hiroshi Iwasaki) antibody was used.

## Analysis of Rad51 filament dissociation kinetics

Anisotropy buffer (30 mM HEPES-KOH [pH 7.5], 100 mM KCl, 10 mM NaCl, 3 mM MgCl$_2$, 1 mM ATP, 1 mM DTT, 5% glycerol) containing 1.5 µMnt of oligo dT (72 mer) with a 5' TAMRA label was supplemented with 0.5 µM Rad51 and incubated at 25℃ for 5 min. Next, a Swi5-Sfr1 variant was added at the indicated concentration and the reaction was incubated at 25℃ for a further 5 min. This solution was transferred into a 0.3 × 0.3 cm quartz cuvette and the fluorescence anisotropy was monitored once per second for 60 s (25℃, excitation 546 nm, emission 575 nm) to confirm filament formation. Next, a 1.0 × 1.0 cm quartz cuvette containing 2 mL of anisotropy buffer was placed into the spectrofluorometer with constant stirring (450 r.p.m) and, after 60 s of measurement, 50 µL of the solution containing Rad51 filaments with or without Swi5-Sfr1 was injected into this cuvette. Fluorescence anisotropy was then monitored once per second for the indicated time. Dissociation rate constants ($k_{off}$) were calculated in KaleidaGraph. Cuvettes were purchased from Hellma Analytics.

## ATPase assay

ATPase buffer (30 mM Tris-Cl [pH 7.5], 100 mM KCl, 20 mM NaCl, 3.5 mM MgCl$_2$, 5% glycerol) containing 10 µMnt PhiX virion DNA was mixed on ice with 5 µM Rad51 and 0.25 µM of a Swi5-Sfr1 variant. Reactions were initiated through the addition of 0.5 mM ATP. Time zero was immediately withdrawn (10 µL) and mixed with 2 µL of 120 mM EDTA to terminate the reaction. Following incubation at 37℃, aliquots were withdrawn at the indicated timepoints and processed as above. Upon

completion of the time course, aliquots were diluted two-fold with water to reduce the concentration of ATP to 0.25 mM. Inorganic phosphate generated by ATP hydrolysis was then detected using a commercial malachite green phosphate detection kit (BioAssay Systems).

## Electrophoretic mobility shift assay

DNA binding buffer (25 mM HEPES-KOH [pH 7.5], 100 mM NaCl, 3.5 mM MgCl$_2$, 1 mM DTT, 5% glycerol) containing 5 µMnt PhiX174 virion DNA or 5 µMnt ApaLI-linearized PhiX RF I DNA (both NEB) was supplemented with the indicated concentration of Swi5-Sfr1 (wild type or mutants) in a 10 µL reaction. Following incubation at 37°C for 15 mins, 2 µL of loading dye was added and 8 µL of the reaction mixture was separated by agarose electrophoresis (0.8% gel in TAE buffer, 2.5 hr 4°C). The gel was stained with SYBR Gold (Thermo Fisher Scientific).

## Quantification and statistical analysis

### DNA damage sensitivity assay

For the clonogenic survival assay, colonies were counted after 3 (rad$^+$) or 4 (rad$^-$) days of growth. The percentage of survival on the control plate (no UV treatment) was set to 100%. The expected number of colonies, based on the number of cells plated, was determined for the plates irradiated with the indicated dose of UV. The number of actual colonies was expressed as a fraction of the expected number, yielding the percentage of cells that survived the treatment. The values from three independent experiments were averaged and plotted, with the standard deviation of these averaged values depicted by error bars. In *Figure 7B*, a one-way ANOVA followed by Tukey's multiple comparisons test was performed. *, $p < 0.05$, n.s., not significant ($p > 0.05$). Further statistical information for *Figure 7B* is listed below.

| Tukey's multiple comparisons test | Mean diff. | 95.00% CI of diff. | Significant? | Summary | Adjusted p value |
|---|---|---|---|---|---|
| WT vs. sfr1d | 23.8 | 13.4 to 34.21 | Yes | ** | 0.001 |
| WT vs. 7A | 10.16 | −0.2475 to 20.57 | No | ns | 0.0547 |
| sfr1d vs. 7A | −13.64 | −24.05 to −3.234 | Yes | * | 0.0163 |

| Test details | Mean 1 | Mean 2 | Mean diff. | SE of diff. | n1 | n2 | Q | DF |
|---|---|---|---|---|---|---|---|---|
| WT vs. sfr1d | 25.14 | 1.337 | 23.8 | 3.392 | 3 | 3 | 9.92 | 6 |
| WT vs. 7A | 25.14 | 14.98 | 10.16 | 3.392 | 3 | 3 | 4.24 | 6 |
| sfr1d vs. 7A | 1.337 | 14.98 | −13.64 | 3.392 | 3 | 3 | 5.69 | 6 |

### Three-strand exchange assay

Following staining with SYBR Gold (Thermo Fisher Scientific), gels were imaged using a LAS4000 mini (GE Healthcare). Densitometric analysis was performed using Multi Gauge software (version 3.2, Fujifilm) exactly as described (*Haruta et al., 2006*; *Kurokawa et al., 2008*). Background signal above the lds, NC and JM bands was subtracted from the corresponding values. The JM value was divided by 1.5 to compensate for the extra signal generated by these three-stranded DNA molecules. The sum of the values was set to 100%, and the percentage of total DNA corresponding to NC or JM was calculated. For the total yield, the percentage of DNA corresponding to NC and JM was combined. The values from three independent experiments were averaged and plotted, with the standard deviation of these averaged values depicted by error bars.

### Analysis of Rad51 filament dissociation kinetics

To set a consistent end-point for the anisotropy graphs (*Figure 6B–E*), data were portrayed for the reactions until they reached a moving average (20 data points) of 0.106, which is equivalent to the value observed for DNA only. Dissociation rate constants ($k_{off}$) were calculated in KaleidaGraph using the following equation:

Anisotropy = (Amplitude of change in anisotropy) x e$^{(-koff \times t)}$ + (Minimum value of anisotropy)

$k_{off}$ values from three independent experiments were averaged and plotted, with the standard deviation of these averaged values depicted by error bars.

## ATPase assay

Absorbance values at 620 nm (A$_{620}$) were obtained using a Nanodrop spectrophotometer (Thermo Fisher Scientific). A$_{620}$ at time zero was subtracted from each value and these values were converted to concentrations of inorganic phosphate through the use of a standard curve. Graphs were plotted with inorganic phosphate concentration (µM) on the y-axis and time (min) on the x-axis. The gradient of the line of best fit was divided by the concentration of Rad51 (µM) to yield $k_{cat}$ values. $k_{cat}$ values from three independent experiments were averaged and plotted, with the standard deviation of these averaged values depicted by error bars.

## Electrophoretic mobility shift assay

Following staining with SYBR Gold (Thermo Fisher Scientific), gels were imaged using a LAS4000 mini (GE Healthcare). Densitometric analysis was performed using Multi Gauge software (version 3.2, Fujifilm). Background signal in a blank lane adjacent to the area where the css or lds band would be was subtracted from the corresponding region of sample lanes. This value was then normalized by the total lane signal and expressed relative to the no protein control, which was set to 100% unbound DNA. The values from three independent experiments were averaged and plotted, with the standard deviation of these averaged values depicted by error bars.

## Acknowledgements

We thank Tomohiro Koizumi for contributing to the early stages of this study; Yumiko Kurokawa and Yuki Ide for help with protein purification; and Ryoji Miyazaki, Hiroyuki Mori and Yoshinori Akiyama for help with the site-specific crosslinking experiments. This study was supported in part by Grants-in-Aid for Scientific Research on Innovative Areas (15H059749 to HI, 18H04626 and 18H05426 to H Takahashi), for Scientific Research (A) (18H03985 to HI), for Young Scientists (B) (17K15061 to BA), for Scientific Research (B) (18H02371 to H Tsubouchi and 19H03160 to YM), for Early-Career Scientists (19K16039 to KI), and a Research Fellowship (17J04051 to NA) from the Japan Society for the Promotion of Science.

## Additional information

### Funding

| Funder | Grant reference number | Author |
| --- | --- | --- |
| Japan Society for the Promotion of Science | 15H059749 | Hiroshi Iwasaki |
| Japan Society for the Promotion of Science | 18H03985 | Hiroshi Iwasaki |
| Japan Society for the Promotion of Science | 18H04626 | Hideo Takahashi |
| Japan Society for the Promotion of Science | 18H05426 | Hideo Takahashi |
| Japan Society for the Promotion of Science | 17K15061 | Bilge Argunhan |
| Japan Society for the Promotion of Science | 18H02371 | Hideo Tsubouchi |
| Japan Society for the Promotion of Science | 19H03160 | Yasuto Murayama |
| Japan Society for the Promotion of Science | 19K16039 | Kentaro Ito |

| Japan Society for the Promotion of Science | 17J04051 | Negar Afshar |
| --- | --- | --- |

The funding sources were not involved in the study design, data collection and interpretation, or the decision to submit this work for publication.

#### Author contributions
Bilge Argunhan, Conceptualization, Resources, Data curation, Formal analysis, Supervision, Funding acquisition, Investigation, Visualization; Masayoshi Sakakura, Conceptualization, Resources, Data curation, Formal analysis, Supervision, Investigation; Negar Afshar, Resources, Formal analysis, Funding acquisition, Validation, Investigation; Misato Kurihara, Takahisa Maki, Resources, Formal analysis, Validation, Investigation; Kentaro Ito, Resources, Formal analysis, Funding acquisition, Validation, Investigation, Methodology; Shuji Kanamaru, Supervision, Validation, Methodology; Yasuto Murayama, Hideo Tsubouchi, Resources, Formal analysis, Supervision, Funding acquisition; Masayuki Takahashi, Formal analysis, Supervision; Hideo Takahashi, Conceptualization, Resources, Formal analysis, Supervision, Funding acquisition, Project administration; Hiroshi Iwasaki, Conceptualization, Formal analysis, Supervision, Funding acquisition, Visualization, Project administration

#### Author ORCIDs
Bilge Argunhan (ID) https://orcid.org/0000-0002-6023-7654
Masayoshi Sakakura (ID) https://orcid.org/0000-0003-3649-3276
Negar Afshar (ID) https://orcid.org/0000-0002-3448-6710
Kentaro Ito (ID) http://orcid.org/0000-0001-9221-5188
Hideo Tsubouchi (ID) http://orcid.org/0000-0003-0814-8432
Masayuki Takahashi (ID) http://orcid.org/0000-0002-1856-5312
Hiroshi Iwasaki (ID) https://orcid.org/0000-0002-0153-6873

#### Decision letter and Author response
Decision letter https://doi.org/10.7554/eLife.52566.sa1
Author response https://doi.org/10.7554/eLife.52566.sa2

## Additional files

#### Supplementary files
- Supplementary file 1. Key Resources Table.
- Transparent reporting form

#### Data availability
Our only dataset is NMR data deposited to BMRB (ID # 27885). All data (except NMR data) generated or analyzed during this study are included in the manuscript and supporting files. Source data files have been provided for Figures 1 (C,D), 5 (D), 6 (B,C,D,E,F,G), 7(B,D) and Figure 6—figure supplement 1(B).

The following dataset was generated:

| Author(s) | Year | Dataset title | Dataset URL | Database and Identifier |
| --- | --- | --- | --- | --- |
| Sakakura M, Kurihara M, Ito K, Maki T, Argunhan B, Iwasaki H, Takahashi H | 2020 | 27885 | http://www.bmrb.wisc.edu/data_library/summary/index.php?bmrbId=27885 | Biological Magnetic Resonance Data Bank, 27885 |

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
