## [Decision Letter]

**Acceptance summary:**

This study illuminates the interaction of the conserved Rad51 paralogs complex Swi5-Sfr1 with Rad51 using biochemical, biophysical, and genetic approaches. The N-terminal region of Sfr1 in an unstructured region that interacts with Rad51 through two independent sites. The authors conduct a series of nice biochemical experiments to define these two binding sites and evaluate their functional importance for in vitro recombination. in vitro elimination of the N-terminus or the mutational inactivation of the Sfr1-Rad51 interaction lead to robust phenotypes. Interestingly in vivo, a phenotype of those mutants is only revealed in the absence of a second Rad51 paralog complex Rad55-Rad57.

**Decision letter after peer review:**

Thank you for submitting your article "Cooperative interactions facilitate stimulation of Rad51 by the Swi5-Sfr1 auxiliary factor complex" for consideration by *eLife*. Your article has been reviewed by three peer reviewers, including Wolf-Dietrich Heyer as the Reviewing Editor and Reviewer #1, and the evaluation has been overseen by Cynthia Wolberger as the Senior Editor.

The reviewers have discussed the reviews with one another and the Reviewing Editor has drafted this decision to help you prepare a revised submission.

Summary:

Argunhan et al. use NMR and biochemical analysis to interrogate the role of Swi5-Sfr1 in modulating the strand exchange activity of Rad51 in *Schizosaccharomyces pombe*. This study illuminates the interaction of the conserved Rad51 paralogs complex with Rad51 using biochemical, biophysical, and genetic approaches. Given the unstructured nature of the Sfr1N sequence, it is challenging to resolve the interaction interface between Sfr1N and Rad51 using high resolution structural techniques such as X-ray crystallography. To address this, the authors have beautifully combined NMR and proximity cross-linking to unravel the interaction between Sfr1N and Rad51. The results show that the N-terminal region of Sfr1 interacts with Rad51 through two independent sites. The authors conduct a series of nice biochemical experiments to define these two binding sites and evaluate their functional importance for in vitro recombination. in vitro elimination of the N-terminus or the mutational inactivation of the Sfr1-Rad51 interaction lead to robust phenotypes. Interestingly in vivo, a phenotype of those mutants is only revealed in the absence of a second Rad51 paralog complex Rad55-Rad57. The manuscript is clearly written. The biochemical, biophysical and genetic experiments are of high technical quality. However, the key finding of the in vivo dependence of the Sfr1-7A phenotype on the absence of Rad55-Rad57 is lacking an explanation or potential mechanism. While it seems clear the Sfr1 N-terminus is able to interact with Rad51, the role of Swi5-Sfr1 DNA binding is downplayed.

Essential revisions:

1) The key observation is that the in vivo phenotype of the Sfr1-7A mutation is revealed only by absence of the Rad55-Rad57 Rad51 paralog complex. What are the mechanisms involved? How does Rad55-Rad57 affect the Swi5-Sfr1 Rad51 (or the Sfr1-Nterm-Rad51) interaction? The key prediction that Rad55-Rad57 interact with Swi5-Sfr1 needs to be addressed experimentally. The functional characterization of this higher order complex is beyond the scope of this manuscript.

2) It is unclear, why the Sfr1-7A mutation would cause a DNA binding defect of the Swi5-Sfr1 complex, based on the model and the other data. Can the authors provide an explanation and/or define a mechanism by which the 7A mutation impacts DNA binding? More discussion and explanation are needed here, but not additional experimentation.

3) The authors conclude that their designated Site 2 has more transient contacts with Site 1 based on the crosslinking results in Figure 4C, but Figure 5B suggests Site 2 has a higher affinity than Site 1. How do the authors explain this discrepancy? Furthermore, how do the authors rule out the possibility that Site 1 and 2 are not one Rad51 interacting unit in 3D space? Based on the crosslinking results and the fact that mutations of either site results in the same reduction in Rad51 stimulation, Site 1 could be important for making contacts with Rad51, while Site 2 acts as a structural scaffold, hence the lack of crosslinks. More discussion is needed here.

4) One caveat of the crosslinking experiment in Figure 4 and Figure 3—figure supplement 1 is that it entails the replacement of an amino acid residue; thus, it could miss out on a critical interaction site. The discrepancy between the crosslink and NMR data on Site 2 could be explained by this. It should be discussed in the text.

5) Mouse Swi5-Sfr1 study showed that the Rad51 interaction site in the Swi5-Sfr1 complex resides in the C-terminus of Swi5. This manuscript and previous paper from the Iwasaki group showed that Swi5-Sfr1C has some stimulatory effect on Rad51, suggesting there might be residual interaction between Swi5-Sfr1C and Rad51. Does Swi5-Sfr1C bind DNA, and how does it compare to the various mutants? What is the proposed mechanism for Swi5-Sfr1C stimulation of Rad51 if it does not interact with Rad51 or stimulate ATPase activity? Could it be through an interaction with DNA? Please provide some consideration and discussion of this point.

6) Figure 6—figure supplement 1A: The DNA binding aspect of the Swi5-Sfr1 was not well addressed experimentally, although the defect in the 7A mutant is apparent. The DNA binding experiments should be run in triplicate and quantified.

7) In the last paragraph of the subsection “Sites 1 and 2 cooperatively facilitate the physical and functional interaction between Swi5-Sfr1 and Rad51”, the authors conclude that unproductive interactions between Swi5-Sfr1 and Rad51 reduce the amount of strand exchange product, but there is not enough evidence to reach this conclusion, especially considering the 3A and 4A mutants bind dsDNA at those concentrations. Thus, increasing Swi5-Sfr1 concentrations could block strand exchange by sequestering the DNA substrate. This possibility should be discussed as well.

---

## [Author Response]

Essential revisions:

*1) The key observation is that the* in vivo *phenotype of the Sfr1-7A mutation is revealed only by absence of the Rad55-Rad57 Rad51 paralog complex. What are the mechanisms involved? How does Rad55-Rad57 affect the Swi5-Sfr1 Rad51 (or the Sfr1-Nterm-Rad51) interaction? The key prediction that Rad55-Rad57 interact with Swi5-Sfr1 needs to be addressed experimentally. The functional characterization of this higher order complex is beyond the scope of this manuscript.*

Previous attempts by us and other to demonstrate Sfr1-Rad51 complex formation by in vivo co-IP was unsuccessful (Akamatsu et al., 2003; Cipiak et al., 2009). Since this approach was unable to capture an established protein-protein interaction, it seemed unlikely that it would allow us to address whether Swi5-Sfr1 and Rad55-Rad57 form a complex together. We did attempt a co-IP experiment to test whether Rad55-Rad57 forms a complex with Swi5-Sfr1 but could not detect any interaction, perhaps due to the very low levels of intracellular protein (we were unable to detect endogenous levels of Rad55 or Sfr1 by western blotting). To circumvent this problem, we partially purified Rad55-Rad57 by overexpression in *S. pombe* (Figure 7—figure supplement 1C, D; see Materials and methods for details). This partially purified protein preparation was then incubated with highly purified Swi5-Sfr1 and Rad57 was immunoprecipitated. This analysis revealed that Swi5-Sfr1 and Rad55-Rad57 form a complex together (Figure 7E). Importantly, Rad51 was not detectable in the immunoprecipitate (Figure 7F). The absence of Rad55-Rad57’s major interacting partner in this immunoprecipitation experiment, combined with the reasonable purity of the complex (Figure 7—figure supplement 1D), implies that the complex formation between Swi5-Sfr1 and Rad55-Rad57 is mediated by a direct interaction. A detailed description of these findings can now be found in the Results (subsection “Swi5-Sfr1 and Rad55-Rad57 form a Rad51-independent complex”) and Discussion (subsection “Rad51 paralogs promote Swi5-Sfr1-dependent DNA repair”, second paragraph).

2) It is unclear, why the Sfr1-7A mutation would cause a DNA binding defect of the Swi5-Sfr1 complex, based on the model and the other data. Can the authors provide an explanation and/or define a mechanism by which the 7A mutation impacts DNA binding? More discussion and explanation are needed here, but not additional experimentation.

The simplest explanation is that mutation of the seven positively charged Lys/Arg residues to Ala in the 7A mutant has neutralised the electrostatic attraction responsible for DNA binding, the role of which, however, remains a mystery. There is evidence to suggest that the DNA binding activity of Swi5-Sfr1 may be unrelated to DNA repair. Our reasoning for this is three-fold. First, the 4A mutant was more defective for DNA binding than the 3A mutant, yet both mutants showed similar DNA damage sensitivity and impairment in Rad51 stimulation in vitro. Second, despite potentiating Rad51 through highly similar mechanisms (filament stabilization and stimulation of ATP hydrolysis, as revealed by Peter Chi’s group), mouse Swi5-Sfr1 is devoid of DNA binding activity (see Supplementary Figure 5 in Tsai et al., 2012). Third, human Swi5-Sfr1 has been implicated in transcriptional regulation (Feng et al., 2013). While it is not currently known whether human Swi5-Sfr1 binds to DNA, this at least raises the possibility that *S. pombe* Swi5-Sfr1 could have a similar role in regulating transcription, and it is easy to imagine that DNA binding could be relevant to such a role. Consistent with this possibility, Cipak et al., 2009, reported that *S. pombe* Swi5-Sfr1 and the XPG-family RNA nuclease Mkt1 (SPAC139.01c) form a complex. Interestingly, Mkt1 was recently shown to function in RNAi-mediated gene silencing and heterochromatin formation (Taglini et al., 2019). These points have now been described in the Discussion (subsection “Cooperative interactions with Rad51 are the primary function of Sfr1N”, last paragraph).

3) The authors conclude that their designated Site 2 has more transient contacts with Site 1 based on the crosslinking results in Figure 4C, but Figure 5B suggests Site 2 has a higher affinity than Site 1. How do the authors explain this discrepancy? Furthermore, how do the authors rule out the possibility that Site 1 and 2 are not one Rad51 interacting unit in 3D space? Based on the crosslinking results and the fact that mutations of either site results in the same reduction in Rad51 stimulation, Site 1 could be important for making contacts with Rad51, while Site 2 acts as a structural scaffold, hence the lack of crosslinks. More discussion is needed here.

Regarding the apparent discrepancy, one mitigating factor is that the crosslinking results in Figure 4C only involve Sfr1N whereas the IP results in Figure 5B involve full-length Swi5-Sfr1. It is possible that, in the context of the full-length complex, Site 2 plays a more important role in the interaction with Rad51 (perhaps in combination with very weak/transient interactions between Rad51 and Swi5-Sfr1C, as discussed in point 4), whereas the interaction between the Sfr1N and Rad51 relies more on contacts with Site 1. The latter point is also supported by our NMR results, where not only more residues in Site 1 showed a reduction in signal intensity (31 vs. 17), but a higher proportion of residues showed a substantial reduction in signal intensity (~45% [14/31] vs. ~24% [4/17]; compare red underlined residues in Figure 3E). Furthermore, as we mentioned in the Discussion of our initial submission (also related to point 3-2), it is possible that Site 1, which is more hydrophobic in nature, was better able to tolerate the presence of the aromatic pBPA, whereas the insertion of pBPA into Site 2 could have hindered its binding to Rad51 (subsection “Cooperative interactions with Rad51 are the primary function of Sfr1N”, third paragraph).

Sites 1 and 2 may comprise a single Rad51 interacting unit in 3D space, and this cannot be excluded by the current data. It is important to note that Site 2 contains far fewer hydrophobic residues than Site 1. This raises the possibility that, even if Sites 1 and 2 act as a single interacting unit in 3D space, they likely occupy different interfaces on Rad51. The suggestion that Site 1 makes direct contact whereas Site 2 acts as a structural scaffold is another plausible explanation for our results. These possibilities have now been described in more detail in the Discussion (subsection “Cooperative interactions with Rad51 are the primary function of Sfr1N”, fifth paragraph).

4) One caveat of the crosslinking experiment in Figure 4 and Figure 3—figure supplement 1 is that it entails the replacement of an amino acid residue; thus, it could miss out on a critical interaction site. The discrepancy between the crosslink and NMR data on Site 2 could be explained by this. It should be discussed in the text.

We completely agree with the possibility that the replacement of a residue central to the Sfr1N-Rad51 interaction with pBPA could itself disrupt the interaction, leading to a negative result in the crosslinking assay. This caveat has now been described in the Discussion (subsection “Cooperative interactions with Rad51 are the primary function of Sfr1N”, third paragraph).

5) Mouse Swi5-Sfr1 study showed that the Rad51 interaction site in the Swi5-Sfr1 complex resides in the C-terminus of Swi5. This manuscript and previous paper from the Iwasaki group showed that Swi5-Sfr1C has some stimulatory effect on Rad51, suggesting there might be residual interaction between Swi5-Sfr1C and Rad51. Does Swi5-Sfr1C bind DNA, and how does it compare to the various mutants? What is the proposed mechanism for Swi5-Sfr1C stimulation of Rad51 if it does not interact with Rad51 or stimulate ATPase activity? Could it be through an interaction with DNA? Please provide some consideration and discussion of this point.

We previously showed that Swi5-Sfr1C does not bind DNA by EMSA (Figure 4A in Kuwabara et al., 2012; also see point 5). In the referenced paper, we demonstrated that Swi5-Sfr1C can promote Rad51-driven DNA strand exchange through stabilization of the Rad51 filament and stimulation of Rad51’s ATPase activity. However, whereas the full-length Swi5-Sfr1 complex could robustly stimulate these aspects of Rad51 at substoichiometric concentrations (Swi5-Sfr1:Rad51, 1:25 or 1:50), 5-10-fold higher concentrations of Swi5-Sfr1C was required to achieve a similar magnitude of stimulation. We therefore concluded that Swi5-Sfr1C must retain some intrinsic level of interaction with Rad51, but this interaction is just too weak/transient to be detected by conventional means (e.g., IP with purified proteins; Figure 4B in Kuwabara et al., 2012). This is also consistent with why Swi5-Sfr1C was only able to stimulate Rad51 when included at higher concentrations. The binding of Swi5-Sfr1C to Rad51 is likely enforced by Sfr1N, which we think functions as an anchor to keep Swi5-Sfr1C in close proximity to Rad51. Based on molecular modelling (Kokabu et al., 2011), we think that Swi5-Sfr1C could fit into the wide groove of the Rad51 filament, thus locking it in the active conformation. These possibilities have now been mentioned in the Discussion (subsection “Cooperative interactions with Rad51 are the primary function of Sfr1N”, first paragraph).

In assays where direct comparisons have been made (three-strand exchange and ATP hydrolysis), the defects of the Swi5-Sfr1-7A mutant were very similar to those of Swi5-Sfr1C. For example, at 0.25 µM of 7A or Swi5-Sfr1C, the total yield (JM + NC) in the three-strand exchange assay was 10.0 ± 4.3 and 4.4% ± 0.7, respectively; this subtle difference was essentially lost at higher concentrations (e.g., at 5 µM: 7A 44.8% ± 1.5, S5S1C 39.0% ± 1.6). In the ATPase assay, the averaged *k_cat_* values were comparable (7A 0.307 min^-1^ ± 0.068, S5S1C 0.306 min^-1^ ± 0.004), but there seemed to be more variation with the 7A complex, perhaps due to some stochasticity conferred by the presence of the disordered N-terminus.

6) Figure 6—figure supplement 1A: The DNA binding aspect of the Swi5-Sfr1 was not well addressed experimentally, although the defect in the 7A mutant is apparent. The DNA binding experiments should be run in triplicate and quantified.

The experiments have now been conducted in triplicate and quantified (Figure 6—figure supplement 1B). The quantified results are consistent with our observation that the 7A mutant is severely defective for DNA binding, the 4A mutant is moderately defective, and the 3A mutant is only mildly defective. This result has now been mentioned in the Results section (subsection “Sites 1 and 2 cooperatively facilitate the physical and functional interaction between Swi5-Sfr1 and Rad51”, last paragraph) and a more detailed analysis has been included in the Discussion (subsection “Cooperative interactions with Rad51 are the primary function of Sfr1N, last paragraph).

7) In the last paragraph of the subsection “Sites 1 and 2 cooperatively facilitate the physical and functional interaction between Swi5-Sfr1 and Rad51”, the authors conclude that unproductive interactions between Swi5-Sfr1 and Rad51 reduce the amount of strand exchange product, but there is not enough evidence to reach this conclusion, especially considering the 3A and 4A mutants bind dsDNA at those concentrations. Thus, increasing Swi5-Sfr1 concentrations could block strand exchange by sequestering the DNA substrate. This possibility should be discussed as well.

Thank you for the suggestion. As mentioned above in point 5, this possibility has now been stated in the Results section (subsection “Sites 1 and 2 cooperatively facilitate the physical and functional interaction between Swi5-Sfr1 and Rad51”, last paragraph).